# RDB2G-Bench: A Comprehensive Benchmark for Automatic Graph Modeling of Relational Databases

**Dongwon Choi, Sunwoo Kim, Juyeon Kim, Kyungho Kim,**
**Geon Lee, Shinhwan Kang, Kijung Shin**
Kim Jaechul Graduate School of AI, KAIST
{cookie000215, kswoo97, juyeonkim, kkyungho,
geonlee0325, shinhwan.kang, kijungs}@kaist.ac.kr

**Myunghwan Kim**
Kumo.AI
myunghwan@kumo.ai

## Abstract

Recent advances have demonstrated the effectiveness of graph-based machine learning on relational databases (RDBs) for predictive tasks. Such approaches require transforming RDBs into graphs, a process we refer to as **RDB-to-graph modeling**, where rows of tables are represented as nodes and foreign-key relationships as edges. Yet, effective modeling of RDBs into graphs remains challenging. Specifically, there exist numerous ways to model RDBs into graphs, and performance on predictive tasks varies significantly depending on the chosen graph model of RDBs. In our analysis, we find that the best-performing graph model can yield up to a 10% higher performance compared to the common heuristic rule for graph modeling, which remains non-trivial to identify. To foster research on intelligent RDB-to-graph modeling, we introduce `RDB2G-Bench`, the first benchmark framework for evaluating such methods. We construct extensive datasets covering **5 real-world RDBs and 12 predictive tasks, resulting in around 50k graph model–performance pairs** for efficient and reproducible evaluations. Thanks to our precomputed datasets, we were able to **benchmark 10 automatic RDB-to-graph modeling methods on the** 12 **tasks about** $380\times$ **faster** than on-the-fly evaluation, which requires repeated GNN training. Our analysis of the datasets and benchmark results reveals key structural patterns affecting graph model effectiveness, along with practical implications for effective graph modeling. Our datasets and code are available at https://github.com/chlehdwon/RDB2G-Bench.

## 1   Introduction

A relational database (RDB) is a collection of data organized into multiple tables connected by shared keys. RDBs enable systematic and efficient management of related information through query languages such as SQL, and have been widely adopted across diverse industries, including finance [18], healthcare [34], and e-commerce [30]. This widespread use has led to the emergence of diverse machine learning applications built upon RDBs.

For machine learning on RDBs, recent studies have explored graph-based approaches [11, 12, 29, 35, 42, 6], which involve modeling RDBs as graphs. Typically, rows of tables are modeled as nodes in the resulting graph, and foreign key (FK) relationships are represented as edges. Graph modeling, where graph neural networks are subsequently applied, effectively captures structural dependencies and leads to improved performance across a range of machine learning tasks [8, 29, 35].

One important consideration in graph-based approaches is the variety of ways to model RDBs as graphs. For example, a single table row representing a user transaction on an item can be modeled either as (1) a node representing the transaction or (2) an edge linking the corresponding user and item. Additionally, one may choose to include only a subset of tables (i.e., nodes) and FK relationships (i.e., edges), excluding those considered less relevant. We call the process of representing RDBs as graphs **RDB-to-graph modeling**, and refer to the resulting graph representation as a **graph model**.

39th Conference on Neural Information Processing Systems (NeurIPS 2025) Track on Datasets and Benchmarks.

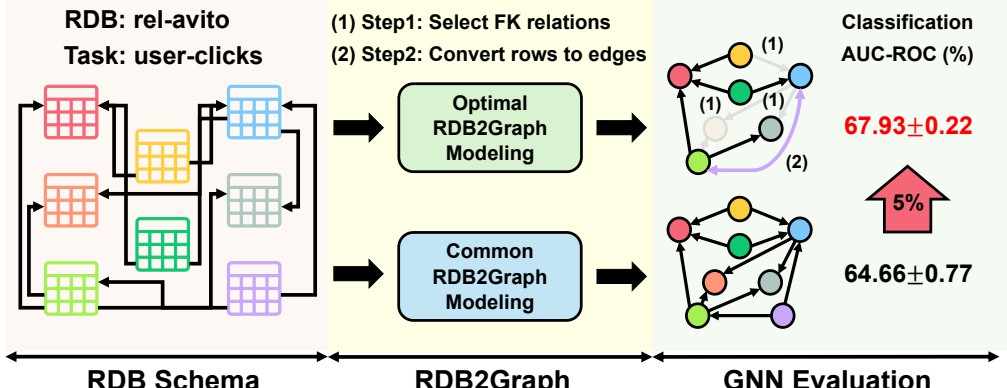

Figure 1: Overview of key concepts. An RDB schema is converted into various network schemas using different RDB-to-graph (RDB2Graph) modeling methods. Graphs are then constructed from these schemas, where graph neural networks (GNNs) are trained and evaluated. In the given example task, optimal modeling yields up to a 5% performance improvement over a widely-used heuristic [11]. Note that the optimal graph model selectively uses tables and foreign key (FK) relations, with table rows modeled as edges, while the heuristic models the entire RDB with all table rows as nodes.

Careful RDB-to-graph modeling is crucial since the empirical effectiveness of graph-based approaches heavily depends on the choice of graph model. For example, our analysis of a graph-based approach [11, 29] on the `rel-avito` dataset shows that modeling the rows of certain tables as edges (rather than nodes) and excluding some FK relations led to a downstream-task performance gain of up to 5% compared to using a commonly-used fixed modeling rule [11], as illustrated in Figure 1.

Despite the importance of intelligent RDB-to-graph modeling, research in this area remains in its early stages, with one main reason being the difficulty of evaluation. First, there are numerous possible graph models (see Figure 2(a) for examples from real-world RDBs), making exhaustive comparisons extremely expensive. Moreover, evaluating even a single graph model is computationally costly, as it typically involves training graph neural networks (GNNs) on a large graph.

To facilitate research on intelligent RDB-to-graph modeling, we introduce `RDB2G-Bench`, the first RDB-to-graph modeling benchmark framework, with the following key contributions:

- **Datasets.** `RDB2G-Bench` provides precomputed performance metrics (training, validation, and test performance and runtime over at least five trials) for **50k graph models** based on 5 real-world RDBs and 12 predictive tasks. They enable researchers to evaluate their modeling methods without training GNNs on the resulting graph models.

- **Benchmarks.** We present the extensive benchmark results of 10 automatic RDB-to-graph modeling methods across 12 tasks. Thanks to our precomputed datasets, we were able to obtain the benchmark evaluation results about **380× faster** than on-the-fly evaluation with repeated GNN training.

- **Observations.** Our analysis of the datasets and benchmark results identifies important factors that impact performance, providing practical insights for effective graph modeling.

Our datasets and code are available at: https://github.com/chlehdwon/RDB2G-Bench.

## 2 Related Works

### 2.1 Relational Deep Learning

Early approaches to relational database (RDB) learning relied on feature engineering [19] (e.g., table joins), transforming an RDB into a single table to leverage tabular machine learning [5, 21, 2, 13]. Recent works on relational deep learning (i.e., deep learning on RDBs) [8, 31, 11, 12, 9] pioneered modeling RDBs as graphs, which facilitate the capture of cross-table dependencies. Building on these graph models, advanced methods [6, 42] have achieved improved predictive performance on RDBs.

## 2.2 Benchmarks on Relational Deep Learning

Early benchmarks (e.g., the CTU relational learning repository [26]) set the stage for evaluating machine learning (ML) models on real-world RDBs. Recent benchmarks (e.g., RDBench [43], RelBench [29], and 4DBInfer [35]) shift the focus to graph-based approaches, which often show improved performance over traditional tabular learning methods [21, 5].

**Distinctive property of our benchmark:** Existing benchmarks provide RDBs and predictive tasks to compare machine learning (ML) methods, which are typically based on the same graph presentation of the RDBs. Our benchmark, RDB2G-Bench, however, is designed to evaluate RDB-to-graph modeling strategies, which provide graph models based on which various modeling methods perform.

Specifically, by offering precomputed performance metrics for 50k graph models, RDB2G-Bench enables comparing various graph modeling strategies without repeated GNN training (see Section 5.2 for efficiency gains).

## 2.3 Graph-based Modeling of RDBs

Traditionally, ML on RDBs has relied heavily on heuristic methods for automating schema transformations. These methods include rule-based relationship mining [1, 40, 25, 23] and conversion of RDBs into single-table formats through predefined aggregation functions [20, 44]. In addition, recent studies have focused on graph modeling, such as mapping table rows to nodes (Row2Node) [29] or to nodes or edges (Row2N/E) [35]. However, heuristic-driven approaches often produce suboptimal RDB representations (see Figure 1 for an example). More recently, AutoG [7] was proposed, leveraging large language models (LLMs) to actively explore effective graph models of RDBs.

Our benchmark, RDB2G-Bench, enables the evaluation of various RDB-to-graph modeling methods, including both heuristic and LLM-based approaches (see Section 5.1 for the ten methods included in the benchmark), ultimately fostering the development of more advanced modeling strategies.

## 3 Dataset Design for RDB2G-Bench

In this section, we introduce the datasets provided in RDB2G-Bench, our benchmark for the RDB-to-graph modeling problem. The problem is defined as follows:

> **Problem Definition.**
>
> **Definition 1** (RDB-to-Graph Modeling).
> - ***Given:*** *A relational database $\mathcal{R}$, and a graph neural network $\mathcal{M}$ for a downstream task $\mathcal{T}$.*
> - *To **Find:** the graph model $\mathcal{G}^*$ of $\mathcal{R}$ that maximizes the performance $\mathcal{P}$ of $\mathcal{M}$ trained and evaluated on $\mathcal{T}$, i.e.,*
>
> $$\mathcal{G}^* = \arg\max_{\mathcal{G} \in \mathcal{F}(\mathcal{R}, \mathcal{T})} \mathcal{P}(\mathcal{M}(\mathcal{G}), \mathcal{T}),$$
>
> *where $\mathcal{F}(\mathcal{R}, \mathcal{T})$ is the set of possible graph models of $\mathcal{R}$ on $\mathcal{T}$, defined in Section 3.1; and example metrics of $\mathcal{P}$ are provided in Section 3.2.*

Each dataset comprises graph models (i.e., graph-structured representations) of a given relational database (RDB) paired with downstream task performance metrics of graph neural networks (GNNs) (denoted by $\mathcal{M}$) trained on those graph models. We first describe the considered design space of graph models (Section 3.1), followed by the procedure for obtaining GNN performance metrics on each graph model (Section 3.2)

**RDBs and downstream predictive tasks.** Our RDB2G-Bench datasets were created based on 5 real-world RDBs and 12 predictive tasks provided in RelBench [29]. The RDBs and tasks, summarized in Figure 2(a), were carefully selected to cover a diverse set of tasks (classification, regression, and recommendation) and domains. Refer to Appendix A.1 for the details of the RDBs and tasks.

## 3.1 Construction of Graph Models

As discussed in Section 1, there are various ways to model each RDB as a graph, and the performance of GNNs depends heavily on the chosen graph model. In `RDB2G-Bench`, we consider the design choices in the following two steps of RDB-to-graph modeling, both of which have a significant impact on the downstream-task performance of GNNs, as shown in Section 4:

- **Step 1:** Selecting which tables and foreign key (FK) relationships to include in the graph model.

- **Step 2:** Selecting how to represent the rows of each table, as either nodes or edges, in the graph.

That is, for each RDB $\mathcal{R}$ (e.g., `rel-avito`), our dataset consists of graph models that result from different combinations of design choices in **Step 1** and **Step 2**, which we denote by $\mathcal{F}(\mathcal{R}, \mathcal{T})$ in Definition 1 (refer to Figure 1 for examples). These graph models are used to train graph neural networks (GNNs) for predictive tasks (e.g., `user-clicks`) defined on the RDB (as described in Section 3.2), and to be used for this purpose, the graph models must satisfy several constraints. First, a valid graph model must select the *task table*, which the considered predictive task is defined on. Second, all selected tables should be connected to the task table via a path whose length does not exceed the number of GNN layers. Violating this constraint would result in some nodes being unreachable during message passing, leading to degraded performance. Third, (the rows of) a table can be modeled as edges only if the table has exactly two FKs and its primary key (PK) is not referenced by any FKs in other tables. Note that, for a table with more than two FKs, hyperedge modeling would be required, which is beyond our scope, and edge modeling of a table whose PK is referenced by FK renders those FK relationships unrepresentable. Also note that these constraints may lead to different graph model spaces for downstream tasks defined on the same RDB. As summarized in Figure 2(a), we constructed about **50k graph models** spanning the aforementioned RDBs and downstream tasks; and evaluated their downstream task performance metrics, as described below.

## 3.2 Collection of Performance Metrics

For each task and each constructed graph model, we collected the following performance metrics (i.e., $\mathcal{P}$ in Definition 1):

- **Predictive performance:** The training, validation, and test performances on the downstream task.

- **Runtime:** The total elapsed time per epoch over at least five trials.

- **Parameter size:** The total number of learnable parameters of predictive GNNs, which depends on the graph model.

Specifically, we collected the performance metrics under the following setups:

**Machines.** All experiments were conducted using NVIDIA RTX A6000 GPUs with 48GB of memory and Intel Xeon Silver 4210/4310 CPUs. Constructing all datasets required about 10,400 GPU hours.

**Predictive GNNs.** As predictive machine learning models, we used GNNs provided by RelBench [29], which leverage PyTorch-Frame [17] to encode each table as input to the GNNs. Specifically, for classification and regression tasks, we employed Heterogeneous GraphSAGE [14] with sum aggregation to update embeddings for the final predictions. For recommendation tasks, we utilized ID-GNN [41]. As the RelBench GNNs do not support edge modeling of tables, we extended them to incorporate edge features and used them when tables are modeled as edges (refer to Appendix A.2). We additionally used three alternative predictive GNNs, as described in Section 4.5.

**Training Details.** Based on the training protocol provided by RelBench [29] (including the training, validation, and test splits), we tuned only the learning rate for each graph model. The other hyperparameters were fixed to the combinations that yielded the best overall performance on 50 randomly sampled graph models. For classification and regression tasks, we selected the learning rate from {0.005, 0.001, 0.0005}, and for recommendation tasks, from {0.001, 0.0005, 0.0001}. The number of training epochs was fixed at 20, which was empirically sufficient for convergence. For reliability, we repeated each experiment with 15 different random seeds for the `rel-f1` and `rel-event` datasets, and with 5 seeds for the remaining datasets. Refer to Appendix A.3 for further details.

**(a)** Summary of the `RDB2G-Bench` datasets, which cover 50k graph models in total.

| RDB | Task Name | Type | # Tables | # Graph Models | Performance Statistics Best | AR2N [29] | Worst |
|---|---|---|---|---|---|---|---|
| `rel-avito` | user-clicks (UC) | classification | 8 | 944 | 67.93 | 64.66 | 60.89 |
| | user-visits (UV) | classification | | 944 | 66.33 | 65.97 | 59.83 |
| | ad-ctr (AC) | regression | | 1304 | 0.039 | 0.040 | 0.044 |
| | user-ad-visit (UAV) | recommendation | | 909 | 3.682 | 3.661 | 0.159 |
| `rel-event` | user-repeat (UR) | classification | 5 | 214 | 82.29 | 77.65 | 63.96 |
| | user-ignore (UI) | classification | | 214 | 82.82 | 82.22 | 74.29 |
| | user-attendance (UA) | regression | | 214 | 0.237 | 0.244 | 0.266 |
| `rel-f1` | driver-dnf (DD) | classification | 9 | 722 | 74.56 | 73.14 | 67.40 |
| | driver-top3 (DT) | classification | | 722 | 81.88 | 78.11 | 75.37 |
| | driver-position (DP) | regression | | 722 | 3.831 | 3.913 | 4.171 |
| `rel-stack` | post-post-related (PPL) | recommendation | 7 | 7979 | 12.04 | 10.82 | 0.006 |
| `rel-trial` | study-outcome (SO) | classification | 15 | 36863 | 70.91 | 68.09 | 62.85 |

**(b)** Visualization of the `RDB2G-Bench` datasets on three predictive tasks.

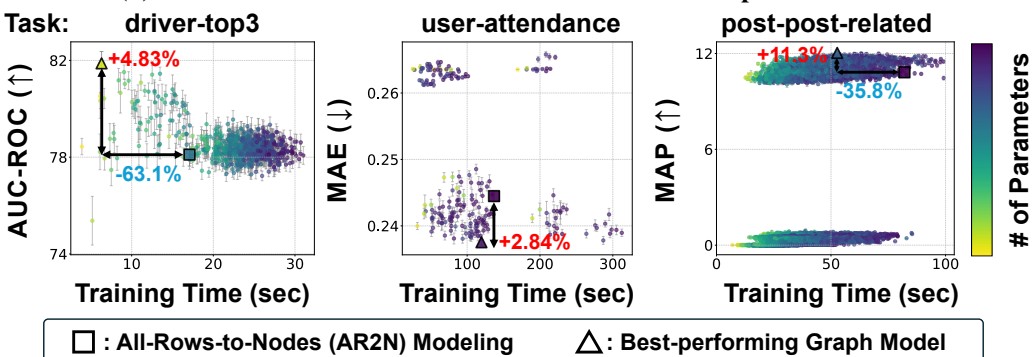

Figure 2: (a) We summarize the RDBs, tasks, and their associated graph models. For each classification, regression, and recommendation task, we collect AUC-ROC (%), MAE, and MAP (%), respectively, on each graph model. For each task, we report the performances on the best graph model, the worst model, and that given by AR2N modeling [29]. (b) For three tasks (`driver-top3`, `user-attendance`, `post-post-related`), we visualize the distribution of performances on the downstream task (Y-axis) across all graph models, along with training time per epoch (X-axis) and the parameter size of the graph neural network (indicated by color). Note that there exist graph models yielding substantial improvements in both performance and efficiency compared to those generated by widely-used AR2N modeling [29].

## 4 Observations from the `RDB2G-Bench` Datasets

In this section, we highlight key observations from our analysis of the constructed `RDB2G-Bench` datasets. Unless otherwise stated, we use the predictive models described in Section 3.2. **For the full analysis results omitted due to space constraints, refer to Appendix B**.

### 4.1 Obs 1. "Finding the best graph models is worthwhile, given their performance benefits."

**Analysis Overview.** We analyze the distribution of downstream-task performances across graph neural networks (GNNs) trained on different graph models of RDBs. We also examine how performance relates to (1) mean GNN training time (per epoch) and (2) the number of GNN parameters.

**Results.** As shown in Figure 2(b), the performance varies significantly across graph models. For example, in the `post-post-related` task, the performance difference spans 11.3%. Notably, for each task, there exist graph models that lead to significantly better performance (e.g., 4.83% improvement in the `driver-top3` task) than the one produced by the widely used all-rows-to-nodes (AR2N) modeling [29], and they often even require shorter training time and fewer parameters.

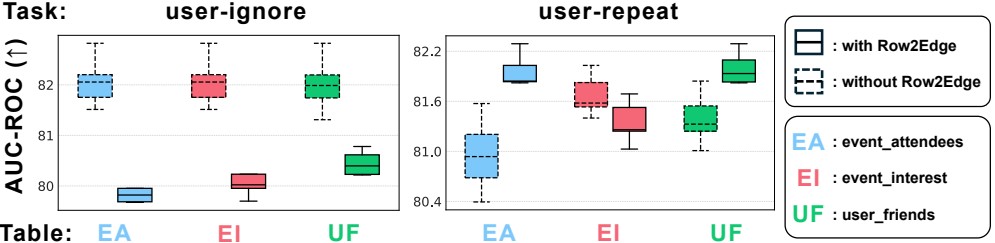

Figure 3: **Modeling table rows as edges (Row2Edge) can be crucial, depending on the task (Obs 2).** EA (event_attendees), EI (event_interest), and UF (user_friends) indicate the tables whose rows can be modeled as edges. Note that Row2Edge modeling improves performance for the user-repeat task, but not for the user-ignore task, even when both are defined on the same RDB.

**Implications.** These results suggest that it is not always beneficial to include all tables and foreign key (FK) relationships in a graph model. Instead, selecting only the most relevant tables can lead to better performance, while also reducing GNN parameters and training time. Moreover, there exist graph models that are both effective and efficient, yet difficult to identify using simple heuristics.

## 4.2 Obs 2. "Modeling table rows as edges can be crucial, depending on the task."

**Analysis Overview.** We compare the two design choices for modeling the rows of a table in an RDB: (1) Row2Node, which always represents table rows as nodes, and (2) Row2Edge, which represents table rows as edges, especially when the table satisfies the constraints discussed in Section 3.1. Specifically, we compare the downstream task performance distributions of the top 10% highest-performing graph models, grouped by design choice.

**Results.** Figure 3 highlights two representative patterns. For the user-ignore task on the rel-event dataset, Row2Edge modeling consistently leads to performance degradation across all tables. In contrast, for the user-repeat task on the same RDB, the Row2Edge modeling of the EA (event_attendees) and UF (user_friends) tables significantly improves AUC-ROC scores.

**Implications.** These findings suggest that even within the same RDB, the benefit of Row2Edge modeling varies significantly across downstream tasks. This variability suggests that no universal rule-of-thumb exists for modeling table rows; the choice should be carefully designed to achieve good performance on the target downstream tasks.

## 4.3 Obs 3. "Top-performing graph models share common substructures."

**Analysis Overview.** We investigate whether top-performing graph models share common substructures by examining the top 1% best-performing graph models for each task.

**Results.** Figure 4 shows a representative case from the user-attendance task on the rel-event dataset. Note that all top-5 best performing graph models include the foreign-key relationship events → users. They also model either the event_attendees table or the event_interest table as edges.

**Implications.** This empirical observation indicates the presence of **common substructures** that are critical for downstream task performance. Finding these structures can be key to identifying effective and efficient graph models.

## 4.4 Obs 4. "Different tasks may require different graph models, even on the same RDB."

**Analysis Overview.** We analyze cross-task performance correlations to examine whether the effectiveness of a graph model in one downstream task implies its effectiveness in others. Specifically, we measure the Spearman rank correlation [33] between performances on tasks within the same RDB.

**Results.** As shown in Figure 5, tasks whose objectives are highly aligned (spec., user-attendance and user-repeat in rel-event)[1] exhibit a high Spearman rank correlation exceeding 0.9. However, most task pairs exhibit low correlations (below 0.4), indicating that a graph model effective for one task may not generalize well to others, even within the same RDB.

---

[1] user-attendance asks how many events each user will attend, and user-repeat asks whether a user will repeat attendance at an event.

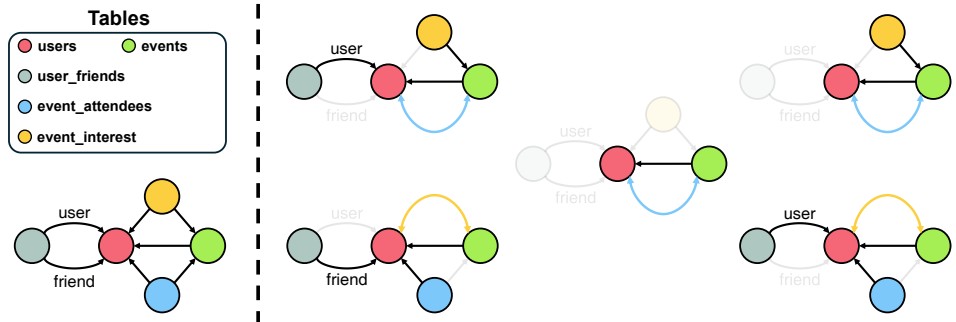

Figure 4: **Top-performing graph models share common substructures (Obs 3).** As shown in their graph models, the top-5 graph models commonly (a) include the foreign-key (FK) relationship events → users and (b) model either the event_attendees table or the event_interest table as edges. Note that the users_friends table has two FKs (user and friend) both referencing the users table.

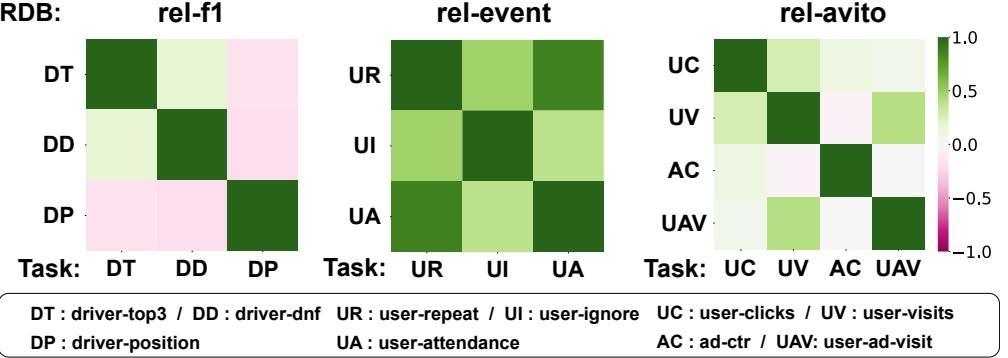

Figure 5: **Different tasks may require different graph models, even on the same RDB (Obs 4).** Spearman correlations between downstream task performances on each RDB (`rel-f1`, `rel-event`, or `rel-avito`) are generally low (below 0.4), except for tasks with closely aligned goals.

**Implications.** These results suggest that an effective RDB-to-graph modeling strategy should account not only for the characteristics of the RDB but also for those of the specific downstream task. Moreover, reusing a graph model is effective only across tasks with closely aligned objectives.

### 4.5   Obs 5. "Effectiveness of graph models generalizes across predictive GNNs."

**Analysis Overview.** We analyze performance correlations across different predictive GNNs applied to the same graph models to examine whether a graph model effective with one GNN (e.g., Heterogeneous GraphSAGE [14]) is also effective with others. To this end, we consider three additional GNNs (GraphSAGE with mean aggregation [14] and GIN [39]) and a graph transformer (GPS [27]) For their details, refer to Appendix A.2. As in the previous analysis, we employ the Spearman rank correlation [33].

**Results.** As shown in Figure 6, the correlations are generally high (above 0.7) in most cases, indicating that the effectiveness of graph models tends to generalize across different GNNs. Notably, the graph transformer also exhibits strong correlations with GNNs, despite their architectural differences.

**Implications.** This cross–GNN generalizability of graph models suggests that RDB-to-graph modeling strategies trained and shown to work well using our benchmark, `RDB2G-Bench`, can be effective across various predictive GNNs, highlighting the broad utility of `RDB2G-Bench`.

## 5   Benchmark Results on the `RDB2G-Bench` Datasets

In this section, we review the benchmark results of ten RDB-to-graph modeling methods on our `RDB2G-Bench` datasets.

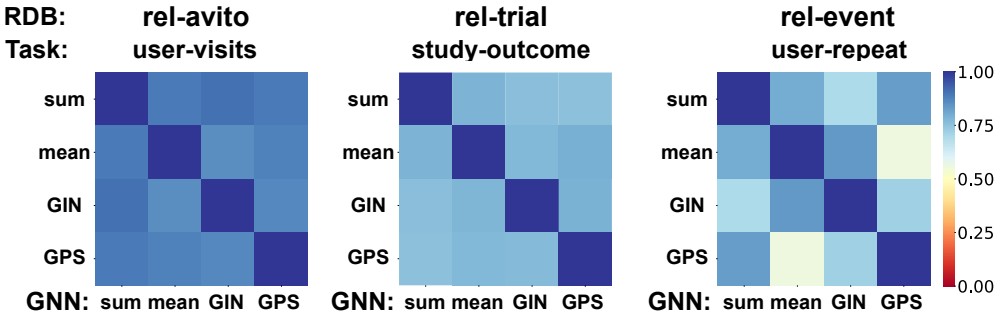

Figure 6: **Effectiveness of graph models generalizes across predictive GNNs (Obs 5).** Spearman correlations between different predictive GNNs—GraphSAGE with sum and mean aggregation (denoted as sum and mean), GIN, and GPS—are generally high (above 0.8) across the three RDBs (`rel-avito`, `rel-trial`, and `rel-event`).

### 5.1 Baselines for RDB-to-Graph Modeling

We benchmark ten RDB-to-graph modeling methods, which are categorized into (a) heuristic-based methods, (b) action-driven search algorithms, and (c) LLM-based approaches. All methods are designed to select a graph model among those in our `RDB2G-Bench` datasets (see Section 3.1 for details on our graph model space).

**Heuristic-Based Methods.** The following heuristic-based methods rely on simple rules to model an RDB as a graph, without explicitly searching for effective graph models:

- **S1. Random [4, 24]:** It randomly samples graph models up to the budget and selects the one with the highest downstream-task performance.
- **S2. All-Rows-to-Nodes (AR2N) [29]:** This widely-used method includes all tables and foreign-key (FK) relationships in the graph, with all table rows modeled as nodes, as shown in Figure 1.

**Action-Based Search Algorithms.** Action-based search algorithms explore and optimize graph models by iteratively modifying them using a predefined set of operations, referred to as *actions*.

We consider the following four actions, which are designed to effectively span our graph model space: (1) `add_fk_edge`: adding an FK relationship between tables to the graph model, (2) `remove_fk_edge`: removing an existing FK relationship, (3) `convert_row_to_edge`: changing the modeling of a table from `Row2Node` to `Row2Edge`, and (4) `convert_edge_to_row`: changing the modeling of a table from `Row2Edge` to `Row2Node`. We build the following six methods based on these actions:

- **S3. Greedy Forward Search (GF):** It starts from the target table and greedily repeats an action (except for `remove_fk_edge`) that yields the greatest improvement to the current graph in terms of downstream-task performance. Recall that tables that are disconnected or distant from the target table are excluded from the graph model, as described in Section 3.1.
- **S4. Greedy Backward Search (GB):** It starts from the AR2N graph, which includes the entire RDB, and greedily repeats an action (except for `add_fk_edge`) that yields the greatest improvement to the current graph.
- **S5. Greedy Local Search (GL):** It starts from a random graph and greedily repeats an action of any type that provides the greatest improvement to the current graph.
- **S6. Evolutionary Algorithm (EA):** It applies evolutionary principles, including *mutation* and *selection*, to iteratively evolve graph models over generations. It randomly applies the predefined actions of any type to mutate the current graph and search for improved ones. Our implementation follows the regularized evolution strategy [28].
- **S7. Bayesian Optimization (BO):** It applies a Bayesian optimization algorithm, specifically BANANAS [37], to the RDB-to-graph modeling task. This approach efficiently explores the graph model space by (a) modeling the function that maps actions to performance and (b) iteratively selecting actions of any type that are likely to improve the graph.
- **S8. Reinforcement Learning (RL):** It applies a controller based on recurrent neural networks [32, 15] and trains it via a policy gradient descent approach [38, 3]. The RL-based predictor learns

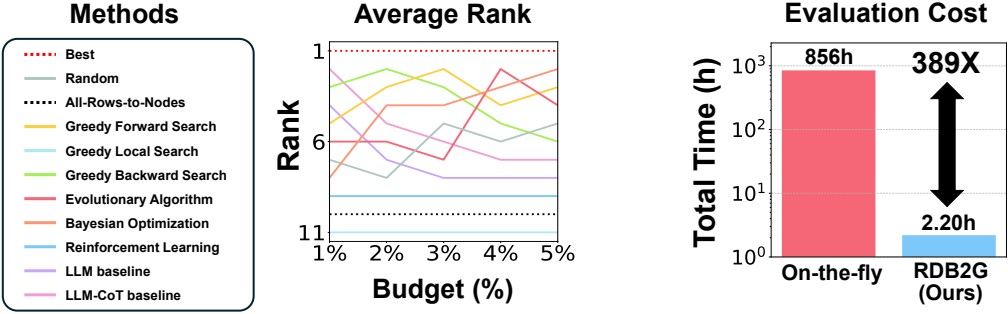

(a) **Rank Averaged over Tasks**  (b) **Evaluation Cost**

Figure 7: (a) Performance ranks of ten RDB-to-graph modeling methods, averaged across 12 predictive tasks. The ranks are computed under varying budget levels, corresponding to the number of graph models evaluated. (b) Comparison of evaluation costs (total elapsed time for all benchmark experiments) between two settings: on-the-fly and `RDB2G-Bench`. `RDB2G-Bench` **speeds up benchmarking by 389×** by eliminating on-the-fly graph model evaluation and associated GNN training.

> to select actions of any type at each step of the RDB-to-graph modeling process to optimize downstream-task performance.

Further details of the action-based search algorithms are provided in Appendix C.

**LLM-Based Baselines.** Inspired by AutoG [7], we developed the following two LLM-based baselines that generate action sequences for RDB-to-graph modeling using LLMs.

- **S9. LLM baseline (LLM):** It directly generates action sequences that specify how a given RDB is expressed as a graph based on the given prompts. At each step, an LLM is provided with a set of candidate actions and selects a sequence of actions expected to be most effective, based on its reasoning capabilities. The key distinction from AutoG lies in the simplified prompt design. Prompts irrelevant due to differences in action spaces are omitted.
- **S10. LLM-CoT baseline (LLM-CoT):** It applies Chain-of-Thought [36] prompt design to encourage complex reasoning in the LLM baseline during the action selection step.

For fair comparisons with the above action-based search algorithms, both baselines operate over the same predefined set of action types. **Claude Sonnet-3.5** is used as the backbone LLM, and details of the prompt designs are provided in Appendix F.

## 5.2  Benchmark Results

For fair comparisons, we evaluate all baselines under a *budget*, which limits the number of graph models whose effectiveness (e.g., ground-truth downstream-task performance) can be measured. We track the effectiveness of resulting graph models as the budget increases up to 5% of the total search space. Full results, omitted here due to space constraints, are provided in Appendix D.

Figure 7a shows the ranks of the baselines, averaged over 12 predictive tasks from `RDB2G-Bench` and 10 independent runs per baseline (3 runs for the LLM-based method). In addition, Table 1 shows the details results on the `rel-f1` dataset. Based on these results, we derive several key observations:

- Most baselines outperform `AR2N` modeling [29] with minimal exploration, suggesting that actively searching for effective graph models is preferable to relying on a fixed modeling rule.
- Greedy methods and `Random` perform comparably to advanced approaches such as `Bayesian Optimization` and `Reinforcement Learning`. Especially, `Greedy Backward` performs best under small budgets, while `Bayesian Optimization` outperforms it as the budget increases. Therefore, the choice between them can be guided by the available computational resources or time constraints.
- Value-based optimization algorithms, such as `Bayesian Optimization` and `Greedy Forward`, typically improve in effectiveness as performance feedback accumulates. However, under limited compute budgets, more complex algorithms, such as `Reinforcement Learning`, or algorithms

Table 1: Performance of ten RDB-to-graph modeling methods on the `rel-f1` dataset under varying budget levels. Refer to Appendix D.1 for results on other datasets.

| Task Name | driver-dnf (AUC-ROC (%) ↑) | | | | | driver-top3 (AUC-ROC (%) ↑) | | | | | driver-position (MAE ↓) | | | | |
|---|---|---|---|---|---|---|---|---|---|---|---|---|---|---|---|
| Methods | Budget (%) | | | | | Budget (%) | | | | | Budget (%) | | | | |
| | 1% | 2% | 3% | 4% | 5% | 1% | 2% | 3% | 4% | 5% | 1% | 2% | 3% | 4% | 5% |
| Best | 74.557 | 74.557 | 74.557 | 74.557 | 74.557 | 81.879 | 81.879 | 81.879 | 81.879 | 81.879 | 3.8311 | 3.8311 | 3.8311 | 3.8311 | 3.8311 |
| Random | 73.225 | 73.450 | 73.592 | 73.745 | 73.755 | 79.627 | 80.165 | 80.425 | 80.438 | 80.604 | 3.8498 | 3.8435 | 3.8420 | 3.8405 | 3.8399 |
| AR2N | 73.140 | 73.140 | 73.140 | 73.140 | 73.140 | 78.106 | 78.106 | 78.106 | 78.106 | 78.106 | 3.9125 | 3.9125 | 3.9125 | 3.9125 | 3.9125 |
| GF | 73.461 | 74.557 | 74.557 | 74.557 | 74.557 | 81.879 | 81.879 | 81.879 | 81.879 | 81.879 | 3.8591 | 3.8409 | 3.8409 | 3.8409 | 3.8409 |
| GB | 73.254 | 74.040 | 74.394 | 74.394 | 74.394 | 80.148 | 80.558 | 80.558 | 80.558 | 80.558 | 3.8437 | 3.8437 | 3.8437 | 3.8437 | 3.8437 |
| GL | 73.023 | 73.645 | 73.726 | 73.775 | 73.775 | 79.172 | 79.292 | 79.299 | 79.299 | 79.299 | 3.8649 | 3.8512 | 3.8500 | 3.8500 | 3.8500 |
| EA | 73.082 | 73.393 | 73.590 | 73.786 | 73.929 | 79.153 | 79.774 | 80.116 | 80.273 | 80.549 | 3.8470 | 3.8451 | 3.8423 | 3.8394 | 3.8394 |
| BO | 73.328 | 73.624 | 73.837 | 73.919 | 74.070 | 79.424 | 79.799 | 80.127 | 80.152 | 80.348 | 3.8520 | 3.8404 | 3.8399 | 3.8394 | 3.8388 |
| RL | 73.332 | 73.526 | 73.602 | 73.657 | 73.986 | 79.043 | 79.385 | 79.439 | 79.906 | 80.084 | 3.8542 | 3.8449 | 3.8417 | 3.8411 | 3.8411 |
| LLM | 73.857 | 73.857 | 73.857 | 73.857 | 73.857 | 80.337 | 80.536 | 80.536 | 80.607 | 80.607 | 3.8534 | 3.8534 | 3.8500 | 3.8500 | 3.8500 |
| LLM-CoT | 73.338 | 73.574 | 73.882 | 73.882 | 73.882 | 78.660 | 78.693 | 78.693 | 78.817 | 78.817 | 3.8437 | 3.8437 | 3.8437 | 3.8437 | 3.8437 |

affected by poor initialization, such as `Greedy Local`, often suffer from unstable exploration, resulting in poor performance.

- In the short run, the LLM-based approach demonstrated strong capabilities in rapidly improving performance through contextual reasoning and effective multi-turn planning. However, it struggled to devise effective long-term action plans, resulting in limited gains as the budget increased. CoT reasoning shows slight performance improvements over the LLM baseline in most cases, which provides potential for using LLMs in RDB-to-graph transformation.

The strong performance of simple methods like greedy approaches indicates significant room for improving RDB-to-graph modeling.

**Efficiency Gain due to** `RDB2G-Bench`**:** Our extensive benchmarking is made efficient by `RDB2G-Bench`, i.e., the precomputed performance metrics for graph models. Without `RDB2G-Bench`, it is inevitable to evaluate graph models on the fly during the search process, requiring repeated training of graph neural networks. As shown in Figure 7b, using `RDB2G-Bench` reduces evaluation time $389\times$ from over 850 hours to just 2.20 hours. `RDB2G-Bench` equally benefits new RDB-to-graph modeling methods, fostering research in this direction.

## 6 Conclusions

This study presents `RDB2G-Bench`, the first benchmark designed to evaluate automatic graph modeling methods for relational databases (RDBs). For `RDB2G-Bench`, we precomputed 50k graph models derived from 5 real-world RDBs and 12 predictive tasks, along with their associated downstream performance, runtime, and parameter size. Analysis of the `RDB2G-Bench` datasets reveals that more data or additional tables do not necessarily improve predictive performance. Instead, selectively using fewer, relevant tables often leads to better results with improved efficiency. Moreover, the best-performing graph models vary by task, emphasizing the need for intelligent modeling strategies that account for the characteristics of both RDBs and predictive tasks. Moreover, our extensive benchmark of ten RDB-to-graph modeling strategies reveals substantial room for improvement in this domain, with LLM-based reasoning showing promising potential, despite its current limitations. `RDB2G-Bench`, whose datasets and code are publicly available at https://github.com/chlehdwon/RDB2G-Bench, significantly accelerates the evaluation of diverse RDB-to-graph modeling strategies, by up to $389\times$, thereby facilitating the development of more advanced techniques.

## Acknowledgements

This work was partly supported by the National Research Foundation of Korea (NRF) grant funded by the Korea government (MSIT) (No. RS-2024-00406985, 20%). This work was partly supported by Institute of Information & Communications Technology Planning & Evaluation (IITP) grant funded by the Korea government (MSIT) (No. RS-2024-00438638, EntireDB2AI: Foundations and Software for Comprehensive Deep Representation Learning and Prediction on Entire Relational Databases, 50%) (No. RS-2024-00457882, AI Research Hub Project, 10%) (RS-2025-02653113, High-Performance Research AI Computing Infrastructure Support at the 2 PFLOPS Scale, 10%) (RS-2019-II190075, Artificial Intelligence Graduate School Program (KAIST), 10%).

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

# A Dataset Detail of the `RDB2G-Bench`

Table 2: Statistics of RELBENCH.

| RDB | Task Name | Type | #Rows of Training Table | | | #Unique Entities | %Train/Test Entity Overlap |
|---|---|---|---|---|---|---|---|
| | | | Train | Validation | Test | | |
| rel-avito | user-clicks | classification | 59,454 | 21,183 | 47,996 | 66,449 | 45.3 |
| | user-visits | classification | 86,619 | 29,979 | 36,129 | 63,405 | 64.6 |
| | ad-ctr | regression | 5,100 | 1,766 | 1,816 | 4,997 | 59.8 |
| | user-ad-visit | recommendation | 86,616 | 29,979 | 36,129 | 63,402 | 64.6 |
| rel-event | user-repeat | classification | 3,842 | 268 | 246 | 1,514 | 11.5 |
| | user-ignore | classification | 19,239 | 4,185 | 4,010 | 9,799 | 21.1 |
| | user-attendance | regression | 19,261 | 2,014 | 2,006 | 9,694 | 14.6 |
| rel-f1 | driver-dnf | classification | 11,411 | 566 | 702 | 821 | 50.0 |
| | driver-top3 | classification | 1,353 | 588 | 726 | 134 | 50.0 |
| | driver-position | regression | 7,453 | 499 | 760 | 826 | 44.6 |
| rel-stack | post-post-related | recommendation | 5,855 | 226 | 258 | 5,924 | 8.5 |
| rel-trial | study-outcome | classification | 11,994 | 960 | 825 | 13,779 | 0.0 |

## A.1 RELBENCH Detail

Our research involves five datasets and twelve tasks from RELBENCH [29], each representing diverse domains and varying scales. A detailed description of each dataset and task is provided below. For more details, please refer to the URLs. Additional statistics are presented in Table 2.

`rel-avito` [2]. Avito is a leading online advertisement platform, providing a marketplace for users to buy and sell a wide variety of products and services, including real estate, vehicles, jobs, and goods. The Avito Context Ad Clicks dataset on Kaggle is part of a competition aimed at predicting whether an ad will be clicked based on contextual information. This dataset includes user searches, ad attributes, and other related data to help build predictive GNNs.

1. `user-visits`: Predict whether each customer will visit more than one Ad in the next 4 days.
2. `user-clicks`: Predict whether each customer will click on more than one Ads in the next 4 days.
3. `ad-ctr`: Assuming the Ad will be clicked in the next 4 days, predict the Click-Through-Rate (CTR) for each Ad.
4. `user-ad-visit`: Predict the list of ads a user will visit in the next 4 days.

`rel-event` [3]. The Event Recommendation database is obtained from user data on a mobile app called Hangtime. This app allows users to keep track of their friends' social plans. The database contains data on user actions, event metadata, and demographic information, as well as users' social relations, which captures how social relations can affect user behavior.

1. `user-repeat`: Predict whether a user will attend an event(by responding yes or maybe) in the next 7 days if they have already attended an event in the last 14 days.
2. `user-ignore`: Predict whether a user will ignore more than 2 event invitations in the next 7 days
3. `user-attendance`: Predict how many events each user will respond yes or maybe in the next seven days.

`rel-f1` [4]. The F1 database tracks all-time Formula 1 racing data and statistics since 1950. It provides detailed information for various stakeholders including drivers, constructors, engine manufacturers, and tyre manufacturers. Highlights include data on all circuits (e.g. geographical details), and full historical data from every season. This includes overall standings, race results, and more specific data like practice sessions, qualifying positions, sprints, and pit stops.

---

[2]https://relbench.stanford.edu/datasets/rel-avito/
[3]https://relbench.stanford.edu/datasets/rel-event/
[4]https://relbench.stanford.edu/datasets/rel-f1/

1. `driver-dnf`: For each driver predict the if they will DNF (did not finish) a race in the next 1 month.

2. `driver-top3`: For each driver predict if they will qualify in the top-3 for a race in the next 1 month.

3. `driver-position`: Predict the average finishing position of each driver all races in the next 2 months.

`rel-stack` [5]. Stack Exchange is a network of question-and-answer websites on topics in diverse fields, each site covering a specific topic, where questions, answers, and users are subject to a reputation award process. The reputation system allows the sites to be self-moderating.

1. `post-post-related`: Predict a list of existing posts that users will link a given post to in the next two years.

`rel-trial` [6]. The clinical trial database is curated from AACT initiative, which consolidates all protocol and results data from studies registered on ClinicalTrials.gov. It offers extensive information about clinical trials, including study designs, participant demographics, intervention details, and outcomes. It is an important resource for health research, policy making, and therapeutic development.

1. `study-outcome`: Predict if the trials will achieve its primary outcome.

## A.2 GNN Implementation

In this section, we demonstrate the implementations of GraphSAGE, GIN, and GPS, including message passing and edge feature extensions for `Row2Edge` modeling. All implementations are based on the PyTorch Geometric (PyG) [10] library.

**GraphSAGE Implementation.** The standard GraphSAGE [14] message passing with **sum aggregation** is defined as:

$$\mathbf{h}_i^{(k)} = \sigma \left( \mathbf{W}^{(k)} \cdot \sum_{j \in \mathcal{N}(i)} \mathbf{h}_j^{(k-1)} + \mathbf{W}_r^{(k)} \cdot \mathbf{h}_i^{(k-1)} \right) \tag{1}$$

Where, $\mathbf{h}_i^{(k)}$ is embedding of node $i$ at layer $k$, $\mathcal{N}(i)$ represents set of neighboring nodes of $i$, and $\sigma$ denotes non-linear activation function (e.g., ReLU).

To extend this formulation to incorporate edge features, $\mathbf{e}_{j \to i}$ for `Row2Edge` [35], we concatenate them with neighbor features. Then, `Row2Edge` message passing becomes as follows (the modified part is highlighted in red):

$$\mathbf{h}_i^{(k)} = \sigma \left( \mathbf{W}^{(k)} \cdot \sum_{j \in \mathcal{N}(i)} \left[ \mathbf{h}_j^{(k-1)} \, \| \, \mathbf{e}_{j \to i} \right] + \mathbf{W}_r^{(k)} \cdot \mathbf{h}_i^{(k-1)} \right) \tag{2}$$

GraphSAGE message passing with **mean aggregation**, which used in Section 4.5 is defined as:

$$\mathbf{h}_i^{(k)} = \sigma \left( \mathbf{W}^{(k)} \cdot \frac{1}{N} \sum_{j \in \mathcal{N}(i)} \mathbf{h}_j^{(k-1)} + \mathbf{W}_r^{(k)} \cdot \mathbf{h}_i^{(k-1)} \right) \tag{3}$$

Where $N$ is defined as the number of neighbors.

**GIN Implementation.** Graph Isomorphism Network (GIN) [39] message passing is defined as:

---

[5] https://relbench.stanford.edu/datasets/rel-stack/
[6] https://relbench.stanford.edu/datasets/rel-trial/

$$\mathbf{h}_i^{(k)} = \mathrm{MLP}^{(k)} \left( (1 + \epsilon^{(k)}) \cdot \mathbf{h}_i^{(k-1)} + \sum_{j \in \mathcal{N}(i)} \mathbf{h}_j^{(k-1)} \right) \tag{4}$$

Here, $\mathbf{h}_i^{(k)}$ denotes the embedding of node $i$ at layer $k$, $\epsilon^{(k)}$ is a trainable scalar parameter, and $\mathrm{MLP}^{(k)}$ is a Multi-Layer Perceptron (MLP) network used to transform aggregated messages. In our implementation, we adopt the following form:

$$\mathrm{MLP}^{(k)}(\cdot) = \mathrm{Linear}(d, 2d) \to \mathrm{ReLU} \to \mathrm{Linear}(2d, d) \tag{5}$$

where $d$ is the hidden dimensionality of node features.

To incorporate edge features $\mathbf{e}_{j \to i}$, we adopt the GINE [16] formulation. Specifically, neighbor embeddings are modulated with their corresponding edge features before aggregation, as follows (the modified part is highlighted in red):

$$\mathbf{h}_i^{(k)} = \mathrm{MLP}^{(k)} \left( (1 + \epsilon^{(k)}) \cdot \mathbf{h}_i^{(k-1)} + \sum_{j \in \mathcal{N}(i)} \mathrm{ReLU}(\mathbf{h}_j^{(k-1)} + \mathbf{e}_{j \to i}) \right) \tag{6}$$

**GPS Implementation.** The General, Powerful, Scalable Graph Transformer (GPS) [27] integrates local message passing with global attention, enabling expressive and scalable graph learning. GPS combines two components at each layer: (1) local neighborhood aggregation, and (2) global message exchange via Transformer-style self-attention.

Formally, for each layer $k$, the message passing is performed in two steps:

$$\mathbf{h}_i^{(k)} = \mathrm{LocalMP}^{(k)} \left( \{ \mathbf{h}_j^{(k-1)} : j \in \mathcal{N}(i) \} \right) + \mathrm{GlobalAttn}^{(k)} \left( \{ \mathbf{h}_j^{(k-1)} : j \in \mathcal{V} \} \right) \tag{7}$$

where $\mathbf{h}_i^{(k)}$ is the embedding of node $i$ at layer $k$, $\mathcal{N}(i)$ denotes the set of neighboring nodes of $i$, and $\mathcal{V}$ is the set of all nodes in the graph.

In our implementation, we employ Equation (1) and Equation (2) for the $\mathrm{LocalMP}^{(k)}$ of `Row2Node` and `Row2Edge` modeling, respectively, to naturally extend our settings to transformer-based architectures. To fit the GPS within a single GPU, we reduce the number of sampled neighbors from 128 to 32.

## A.3 Hyperparameter Details

Table 3 shows our learning rate choices, which are only tuned for each task. For the other hyperparameter, all tasks used two-layer GNNs with a batch size of 512, 128 dimensions, and the Adam [22] optimizer.

Specifically, neighbor sampling counts for subgraph extraction are set to 128 when all tables are represented as nodes, and 12 when some tables are represented as edges. A lower sampling count is employed when tables are modeled as edges, since the two-layer GNN effectively captures paths longer than two hops, resulting in a significantly larger graph scale that cannot be processed by a single GPU.

Table 3: The learning rate settings tuned for each task.

| RDB | Task name | Type | Learning Rate |
|---|---|---|---|
| rel-avito | user-clicks | classification | 0.001 |
| | user-visits | classification | 0.001 |
| | ad-ctr | regression | 0.0005 |
| | user-ad-visit | recommendation | 0.001 |
| rel-event | user-repeat | classification | 0.005 |
| | user-ignore | classification | 0.005 |
| | user-attendance | regression | 0.005 |
| rel-f1 | driver-dnf | classification | 0.005 |
| | driver-top3 | classification | 0.005 |
| | driver-position | regression | 0.0005 |
| rel-stack | post-post-related | recommendation | 0.0005 |
| rel-trial | study-outcome | classification | 0.0005 |

# B  Additional Results regarding Our Observations on the `RDB2G-Bench` Datasets

## B.1  Additional Results regarding Obs 1.

Figure 8 complements Section 4.1 by presenting the performance distribution for the remaining 9 tasks. The results for `driver-top3`, `user-attendance`, and `post-post-related` are visualized in Figure 2 (b) in the main paper.

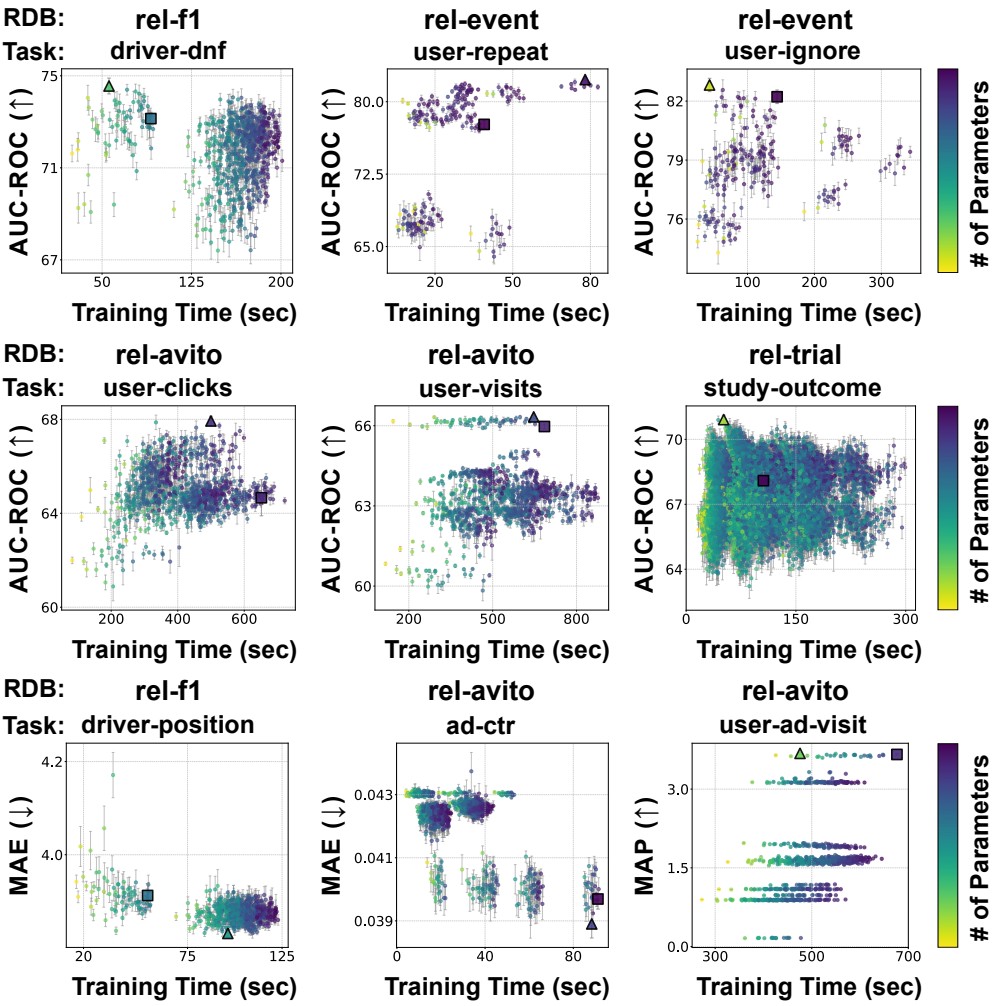

Figure 8: For the nine remaining tasks, we visualize the distribution of performances (Y-axis) across all graph models, along with training time (X-axis) and GNN parameter size (indicated by color). Note that there exist graph models yielding substantial improvements in both performance and efficiency compared to those generated by widely-used AR2N modeling [29].

## B.2 Additional Results regarding Obs 2.

In this section, we provide additional experimental results supporting Section 4.2, which examines how modeling table rows can lead to markedly different outcomes depending on the specific downstream task. Figures 9 - 13 show the results on the remaining 10 tasks. The results for `user-repeat` and `user-ignore` are visualized in Figure 3 of the main paper.

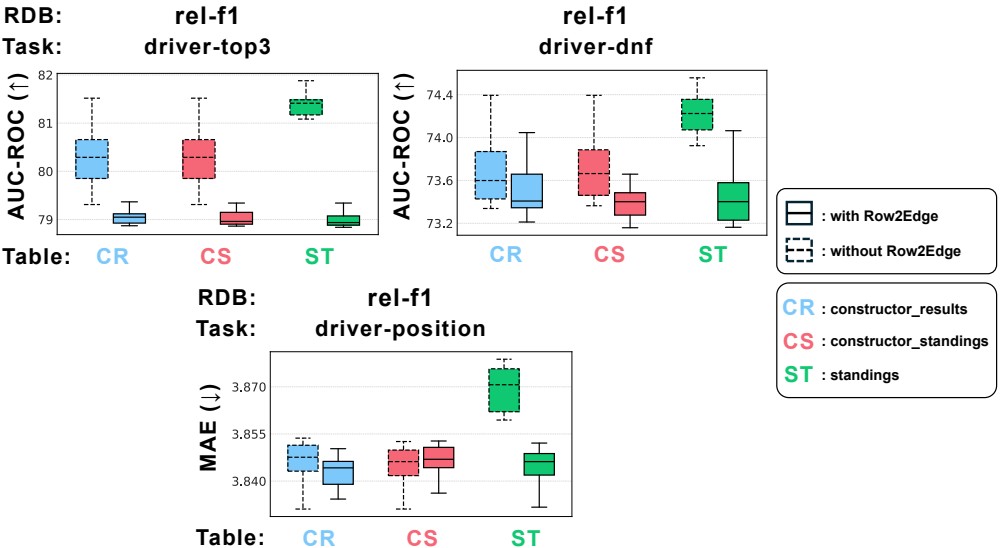

Figure 9: **Additional results on** `rel-f1` **regarding Obs 2.**

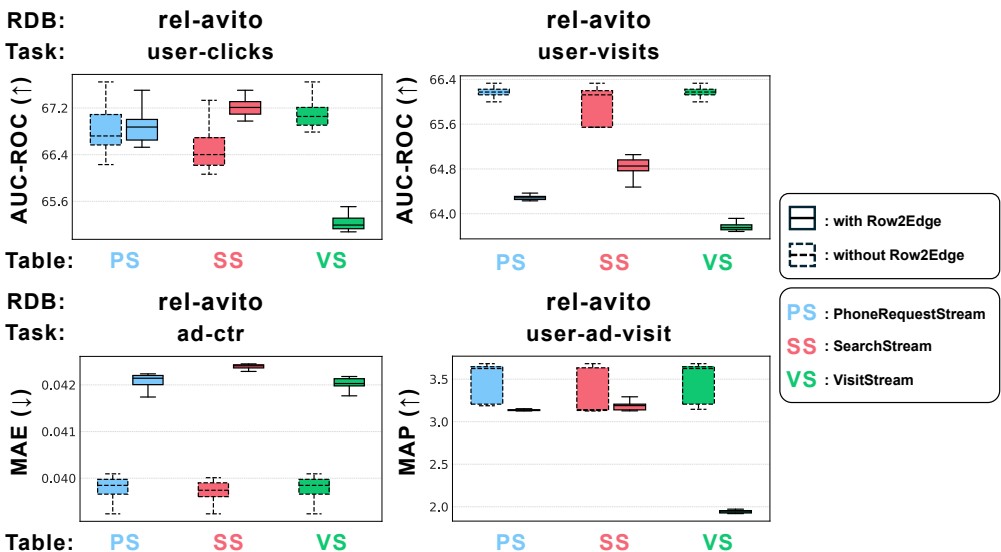

Figure 10: **Additional results on** `rel-avito` **regarding Obs 2.**

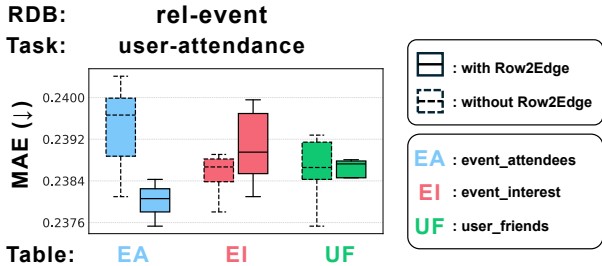

Figure 11: **Additional results on** `rel-event` **regarding Obs 2.**

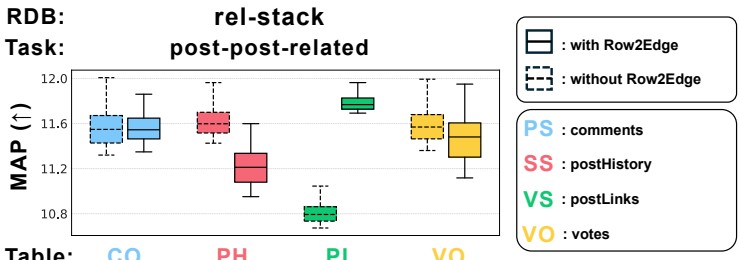

Figure 12: **Additional results on** `rel-stack` **regarding Obs 2.**

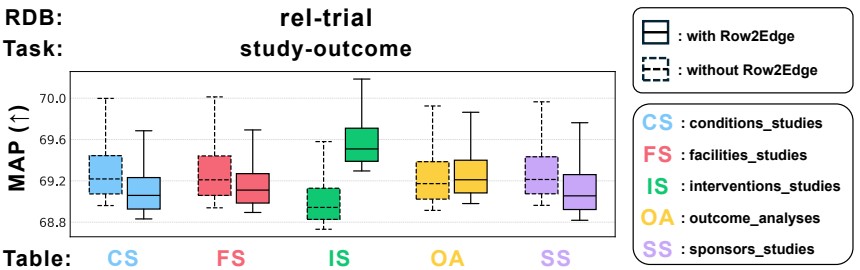

Figure 13: **Additional results on** `rel-trial` **regarding Obs 2.**

## B.3   Additional Results regarding Obs 3.

In this section, we present additional results to supplement Section 4.3. Figures 14 - 24 provide the case study results on the remaining 11 tasks. The case study result for `user-attendance` is provided in Figure 4 in the main paper.

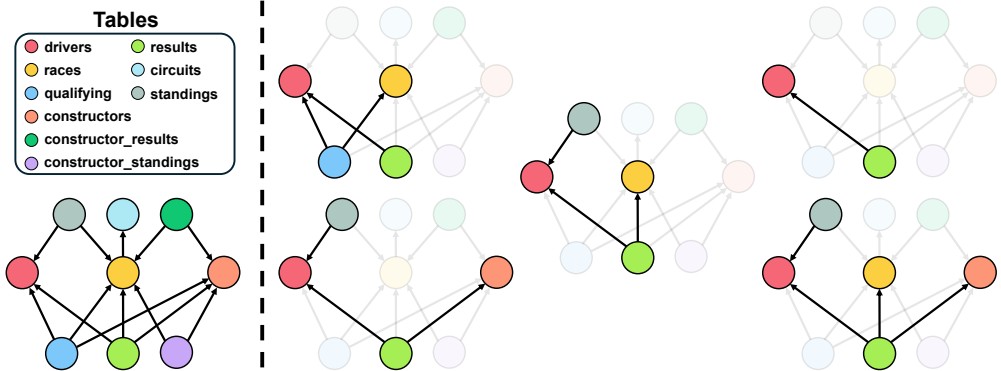

Figure 14: **Top performing graph models for** `driver-top3`.

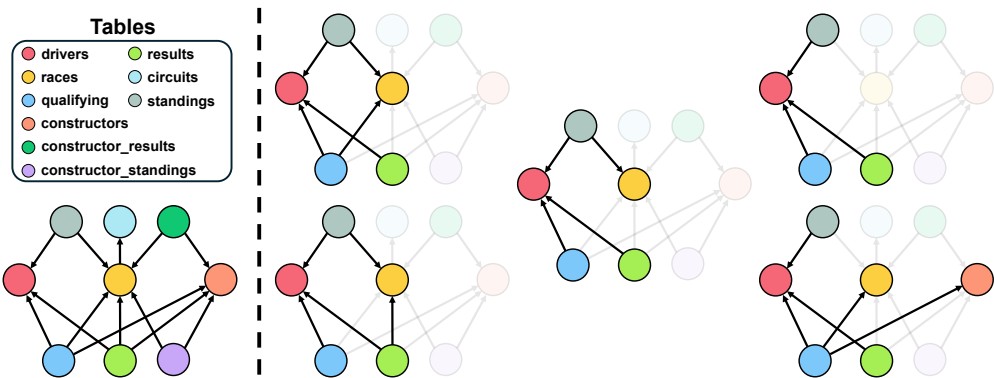

Figure 15: **Top performing graph models for** `driver-dnf`.

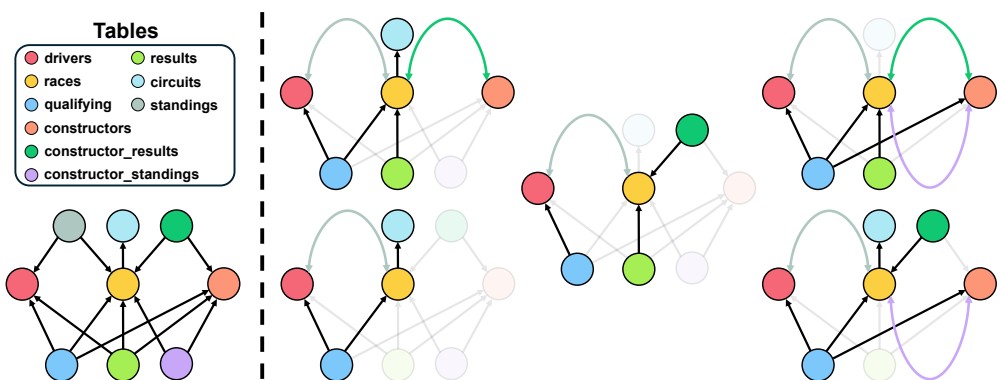

Figure 16: **Top performing graph models for** `driver-position`.

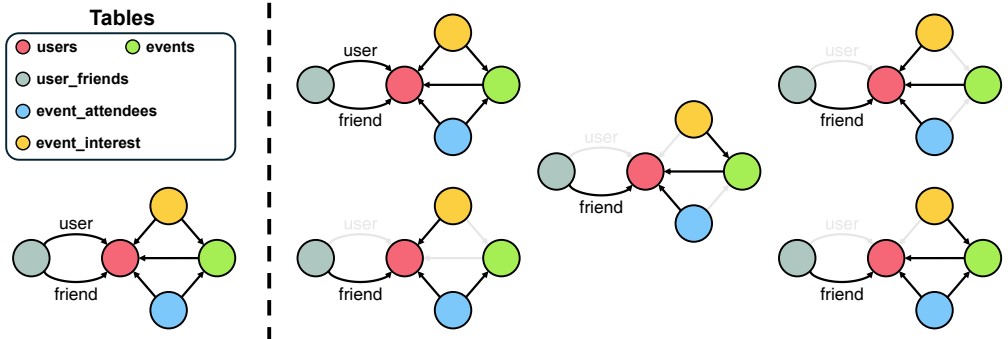

Figure 17: **Top performing graph models for** `user-ignore`. Note that the users_friends table has two FKs (`user` and `friend`) both referencing the users table.

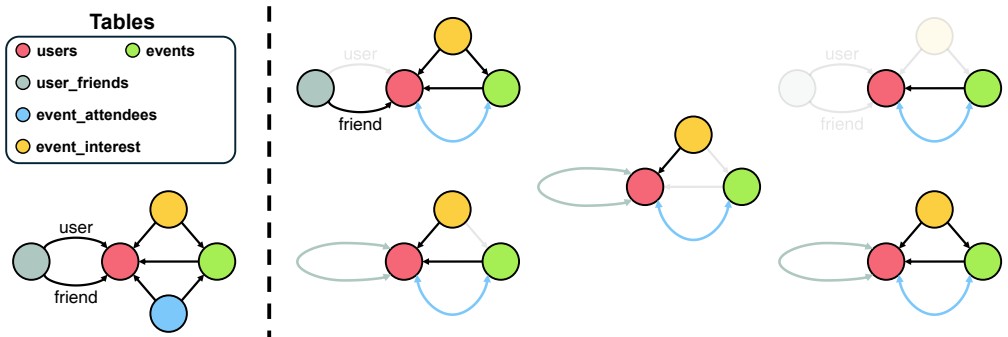

Figure 18: **Top performing graph models for** `user-repeat`. Note that the users_friends table has two FKs (`user` and `friend`) both referencing the users table.

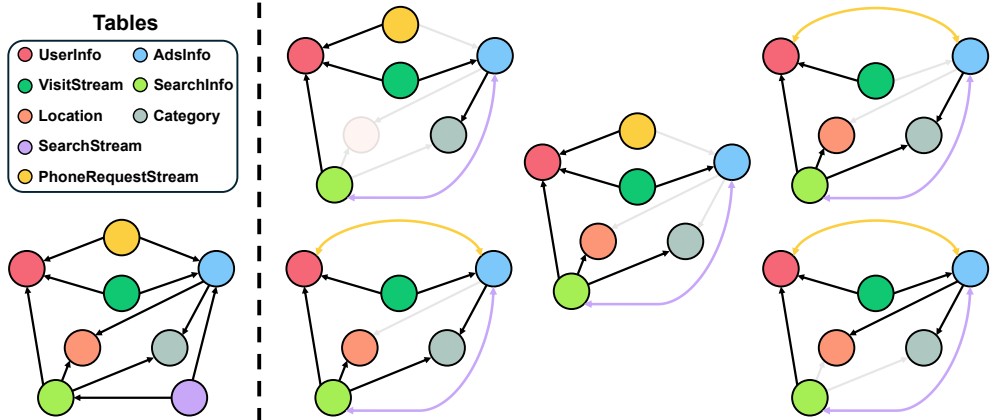

Figure 19: **Top performing graph models for** `user-clicks`.

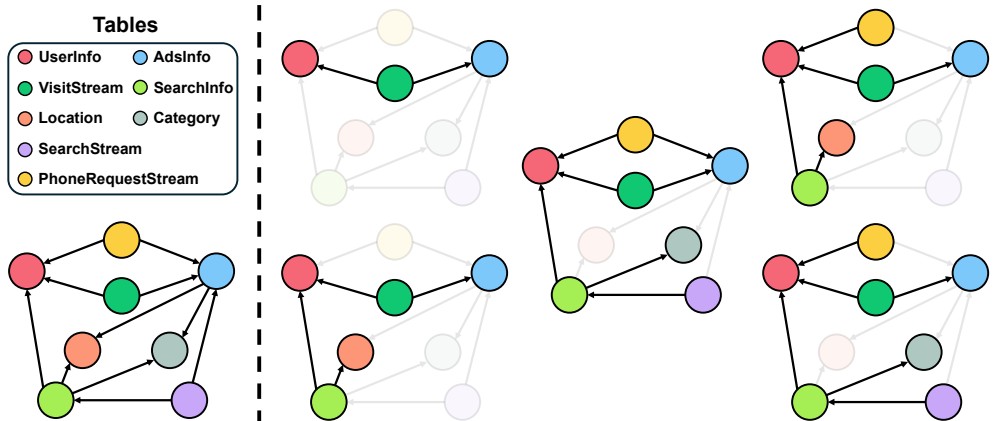

Figure 20: **Top performing graph models for** `user-visits`.

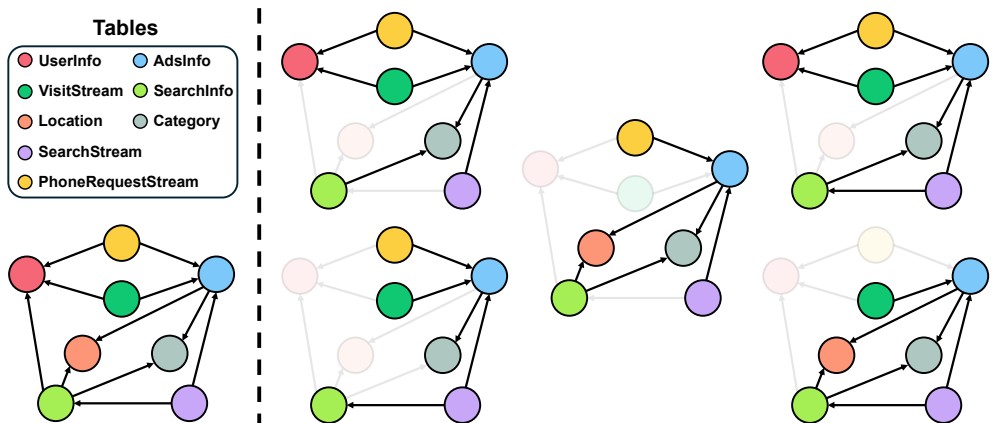

Figure 21: **Top performing graph models for** `ad-ctr`.

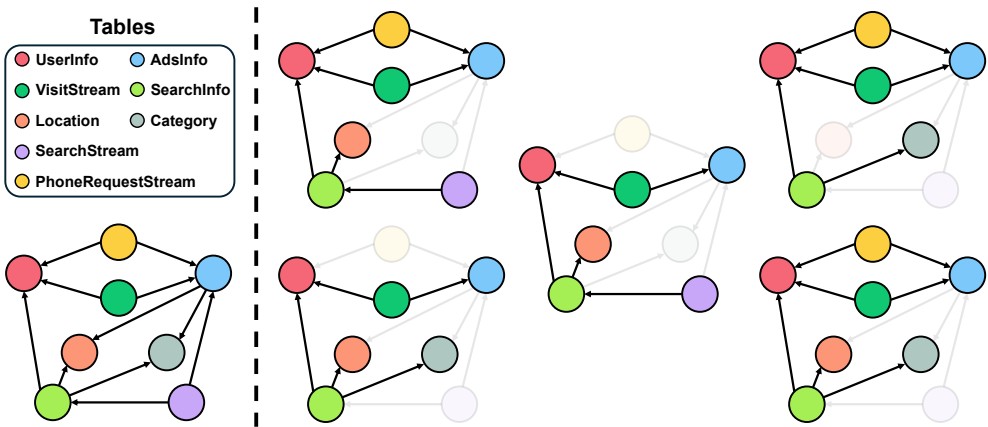

Figure 22: **Top performing graph models for** `user-ad-visit`.

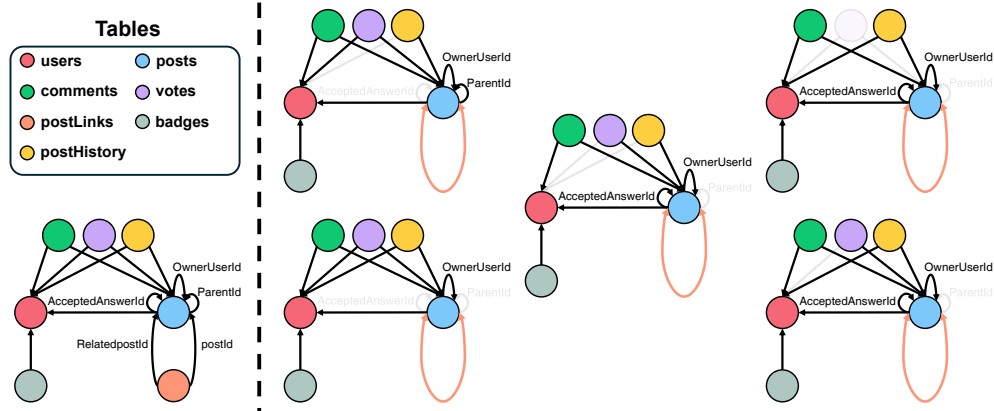

Figure 23: **Top performing graph models for** `post-post-related`**.** Note that the postLinks table has two FKs (`RelatedpostId` and `postId`) both referencing the posts table, and the posts table has three FKs (`AcceptedAnswerId`, `OwnerUserId`, and `ParentId`) all referencing the table itself.

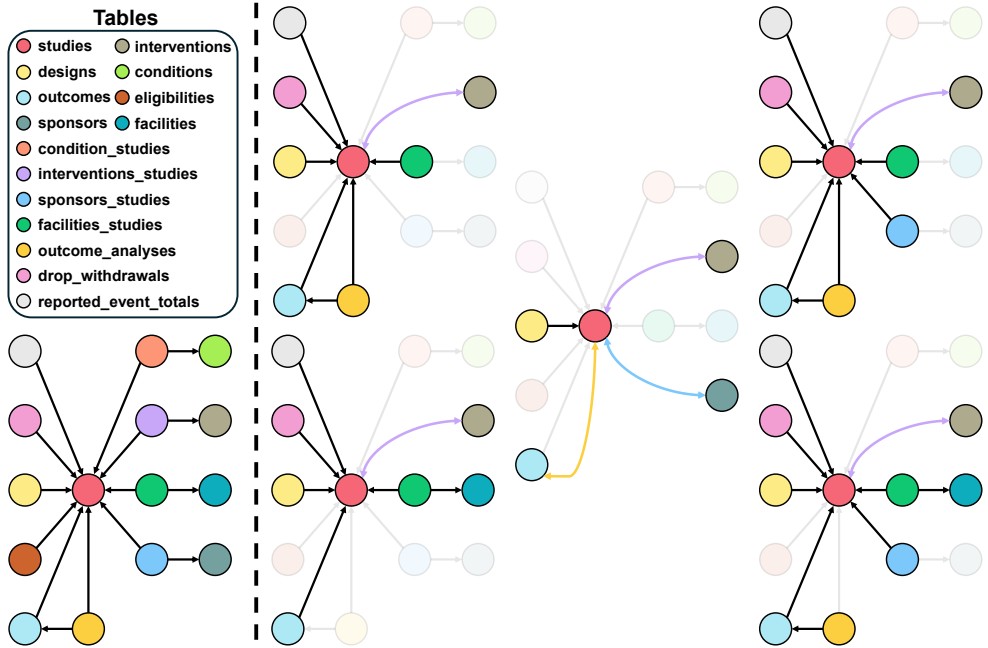

Figure 24: **Top performing graph models for** `study-outcome`**.**

### B.4 Additional Results regarding Obs 5.

In this section, we provide extra details and results to supplement Section 4.5. Given our computational constraints, we examine only the correlations for the top- and bottom-5% graph models—ranked by GraphSAGE (sum aggregation)—across three new predictors (i.e., graph neural networks): GraphSAGE (mean aggregation), GIN, and GPS.

As shown in Figure 25, while cross-GNN correlations are relatively low for the two tasks on the `rel-f1` dataset (`driver-top3` and `driver-position`) due to high variances in the performances, for the other tasks, cross-GNN correlations are high. The results for `user-repeat`, `user-visits`, and `user-ad-visit` are presented in Figure 6 in the main paper.

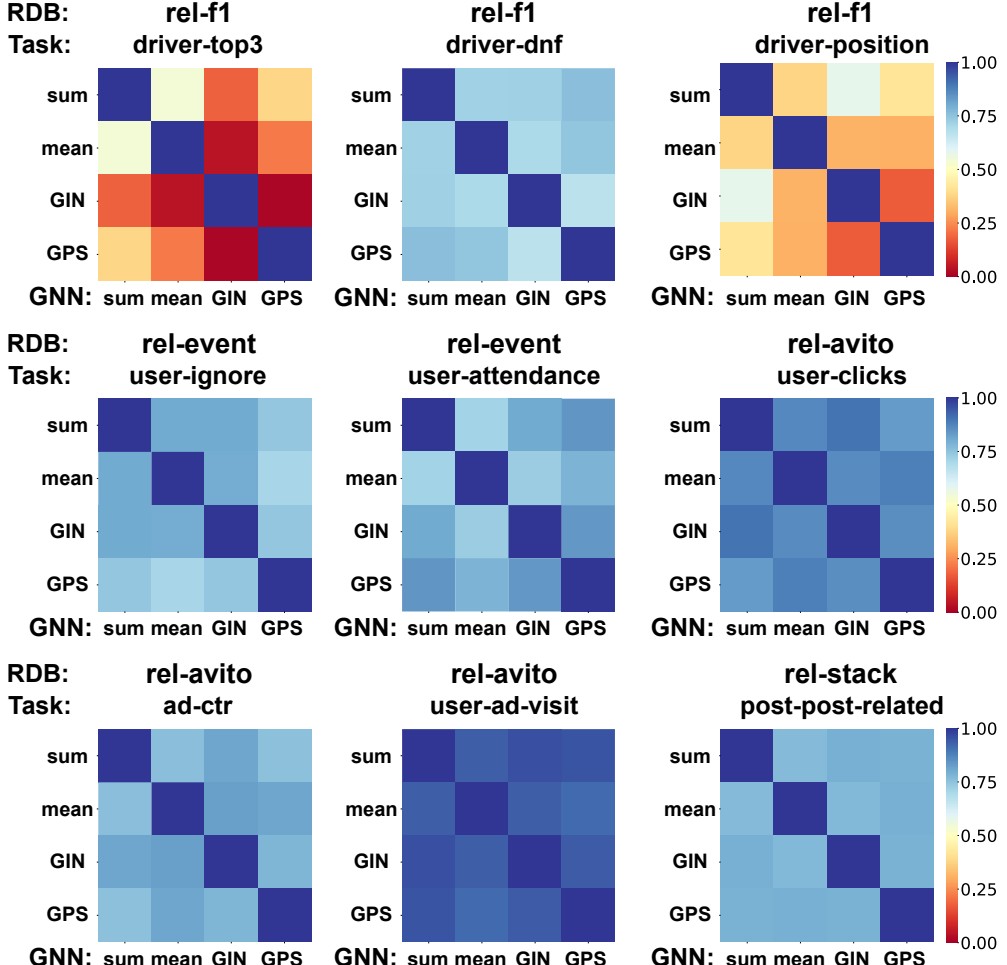

Figure 25: **Additional results regarding Obs 5.** Spearman correlations between different predictive GNNs—GraphSAGE with sum and mean aggregation (denoted as sum and mean), GIN, and GPS—are generally high (above 0.7) across the remaining tasks.

## C  Benchmark Detail of the `RDB2G-Bench`

### C.1  Implementation Details of Baselines

In this section, we provide implementation details for the action-based search algorithms: Evolutionary Algorithm (EA), Bayesian Optimization (BO), and Reinforcement Learning (RL).

**Evolutionary Algorithm.**    The Evolutionary Algorithm (EA) baseline employs a regularized evolutionary search strategy. The algorithm initializes a population by randomly sampling graph

configurations and iteratively evolves this population through mutation and selection processes. At each iteration, a subset of individuals is selected through tournament selection, and mutations are applied using defined micro-actions to generate offspring. The oldest individual in the population is replaced by the best-performing offspring, maintaining diversity and continuous exploration.

Its key hyperparameter settings are summarized below:

- **Population Size**: $\min(10, \text{budget})$
- **Tournament Size**: $\min(10, \text{budget})$
- **Max Iterations**: 1000

**Bayesian Optimization.** The Bayesian Optimization (BO) baseline employs an iterative search leveraging a surrogate Multi-Layer Perceptron (MLP) based on the BANANAS [37] implementation. The input embeddings represent graph structures converted into fixed-sized arrays based on selected edges.

The loss function follows the original BANANAS formulation:

$$L = \text{mean} \left| \frac{y_{\text{pred}} - y_{\text{lb}}}{y_{\text{true}} - y_{\text{lb}}} - 1 \right| \tag{8}$$

Initial embeddings are obtained through random sampling, and subsequent selections are guided by Expected Improvement, computed via Monte Carlo sampling with 50 samples. This iterative process continues until a defined budget limit or a maximum of 100 iterations is reached.

Its key hyperparameter settings are summarized below:

- **Surrogate Model**: MLP (2 hidden layers, each with 32 units)
- **Dropout Rate**: 0.1
- **Optimizer**: Adam
- **Learning Rate**: 0.001
- **Epochs per Iteration**: 50
- **Batch Size**: 32
- **Initial Sampling Size**: $\min(10, \text{budget})$
- **EI MC Samples**: 50
- **Max Iterations**: 100

**Reinforcement Learning.** The Reinforcement Learning (RL) baseline utilizes a policy gradient approach with an RNN-based controller for the search process. The controller is implemented as a one-layer LSTM [15] with 32 hidden units. Input state embeddings are derived from the current graph structures encoded as fixed-size vectors.

Each training episode consists of up to five steps, during which the controller selects actions based on the policy generated from the RNN outputs. Rewards are computed based on performance improvements between sequential states, using a discount factor of 0.99. Training runs for up to 50 episodes or until a predefined evaluation budget is reached. Its key hyperparameter settings are summarized below:

- **Controller Model**: LSTM (1 layer, 32 hidden units)
- **Optimizer**: Adam
- **Learning Rate**: 0.005
- **Max Steps per Episode**: 5
- **Discount Factor** ($\gamma$): 0.99
- **Episodes**: 50

# D  Additional Results regarding Benchmark on the `RDB2G-Bench` Datasets

## D.1  Performance Results on the `RDB2G-Bench` Datasets

In this section, we provide additional performance details of ten RDB-to-graph modeling methods, supplementing the summary presented in Table 1 and Figure 7a in the main paper. Details are presented in Tables 4 - 6 and Figure 26.

Table 4: Performance of ten RDB-to-graph modeling methods on the `rel-event` dataset under varying budget levels.

| Task Name | user-ignore (AUC-ROC (%) ↑) | | | | | user-repeat (AUC-ROC (%) ↑) | | | | | user-attendance (MSE ↓) | | | | |
|---|---|---|---|---|---|---|---|---|---|---|---|---|---|---|---|
| **Methods** | **Budget (%)** | | | | | **Budget (%)** | | | | | **Budget (%)** | | | | |
| | **1%** | **2%** | **3%** | **4%** | **5%** | **1%** | **2%** | **3%** | **4%** | **5%** | **1%** | **2%** | **3%** | **4%** | **5%** |
| Best | 82.823 | 82.823 | 82.823 | 82.823 | 82.823 | 82.291 | 82.291 | 82.291 | 82.291 | 82.291 | 0.2375 | 0.2375 | 0.2375 | 0.2375 | 0.2375 |
| Random | 79.543 | 80.797 | 81.427 | 81.548 | 81.572 | 78.034 | 79.588 | 81.280 | 81.430 | 81.430 | 0.2446 | 0.2423 | 0.2396 | 0.2391 | 0.2389 |
| AR2N | 82.222 | 82.222 | 82.222 | 82.222 | 82.222 | 77.651 | 77.651 | 77.651 | 77.651 | 77.651 | 0.2445 | 0.2445 | 0.2445 | 0.2445 | 0.2445 |
| GF | 77.655 | 78.213 | 78.233 | 78.945 | 80.931 | 77.789 | 79.163 | 80.348 | 80.508 | 80.981 | 0.2447 | 0.2423 | 0.2399 | 0.2399 | 0.2385 |
| GB | 82.222 | 82.222 | 82.222 | 82.222 | 82.222 | 78.748 | 80.469 | 80.744 | 81.018 | 81.258 | 0.2432 | 0.2414 | 0.2409 | 0.2407 | 0.2400 |
| GL | 79.256 | 79.809 | 80.534 | 80.809 | 81.120 | 79.158 | 79.558 | 80.481 | 80.739 | 0.2423 | 0.2418 | 0.2415 | 0.2398 | 0.2391 | |
| EA | 79.939 | 80.523 | 81.011 | 81.299 | 81.482 | 78.989 | 80.739 | 81.249 | 81.313 | 81.333 | 0.2425 | 0.2404 | 0.2395 | 0.2390 | 0.2389 |
| BO | 78.787 | 80.550 | 81.015 | 81.162 | 81.288 | 77.516 | 80.360 | 80.831 | 81.120 | 81.424 | 0.2449 | 0.2403 | 0.2396 | 0.2394 | 0.2392 |
| RL | 79.474 | 80.376 | 80.527 | 80.719 | 80.963 | 79.341 | 80.703 | 80.930 | 81.040 | 81.094 | 0.2474 | 0.2421 | 0.2412 | 0.2410 | 0.2409 |
| LLM | 80.541 | 80.541 | 81.146 | 81.751 | 81.751 | 78.841 | 79.537 | 80.857 | 80.941 | 80.941 | 0.2400 | 0.2397 | 0.2397 | 0.2397 | 0.2397 |
| LLM-CoT | 82.222 | 82.222 | 82.222 | 82.222 | 82.222 | 81.259 | 81.259 | 81.259 | 81.259 | 81.259 | 0.2394 | 0.2394 | 0.2387 | 0.2387 | 0.2387 |

Table 5: Performance of ten RDB-to-graph modeling methods on the `rel-avito` dataset under varying budget levels.

| Task Name | user-clicks (AUC-ROC (%) ↑) | | | | | user-visits (AUC-ROC (%) ↑) | | | | |
|---|---|---|---|---|---|---|---|---|---|---|
| **Methods** | **Budget (%)** | | | | | **Budget (%)** | | | | |
| | **1%** | **2%** | **3%** | **4%** | **5%** | **1%** | **2%** | **3%** | **4%** | **5%** |
| Best | 67.931 | 67.931 | 67.931 | 67.931 | 67.931 | 66.332 | 66.332 | 66.332 | 66.332 | 66.332 |
| Random | 66.594 | 66.856 | 67.098 | 67.151 | 67.302 | 64.444 | 65.169 | 65.521 | 65.708 | 65.877 |
| AR2N | 64.660 | 64.660 | 64.660 | 64.660 | 64.660 | 65.971 | 65.971 | 65.971 | 65.971 | 65.971 |
| GF | 66.978 | 67.096 | 67.096 | 67.096 | 67.096 | 66.318 | 66.318 | 66.318 | 66.318 | 66.318 |
| GB | 66.561 | 66.791 | 66.791 | 66.791 | 66.791 | 66.332 | 66.332 | 66.332 | 66.332 | 66.332 |
| GL | 65.770 | 66.183 | 66.468 | 66.468 | 66.483 | 64.700 | 65.090 | 65.118 | 65.124 | 65.124 |
| EA | 66.778 | 66.944 | 66.958 | 67.204 | 67.232 | 65.352 | 65.397 | 66.047 | 66.116 | 66.122 |
| BO | 66.388 | 67.069 | 67.170 | 67.442 | 67.584 | 64.823 | 65.004 | 65.380 | 65.924 | 66.191 |
| RL | 66.341 | 66.771 | 66.801 | 66.902 | 66.983 | 63.840 | 64.228 | 64.379 | 64.963 | 65.249 |
| LLM | 66.066 | 66.308 | 66.648 | 67.092 | 67.323 | 66.246 | 66.246 | 66.246 | 66.246 | 66.246 |
| LLM-CoT | 66.597 | 66.597 | 66.597 | 66.597 | 66.597 | 66.246 | 66.246 | 66.246 | 66.246 | 66.246 |

| Task Name | ad-ctr (MSE ↓) | | | | | user-ad-visit (MAP (%) ↑) | | | | |
|---|---|---|---|---|---|---|---|---|---|---|
| **Methods** | **Budget (%)** | | | | | **Budget (%)** | | | | |
| | **1%** | **2%** | **3%** | **4%** | **5%** | **1%** | **2%** | **3%** | **4%** | **5%** |
| Best | 0.0389 | 0.0389 | 0.0389 | 0.0389 | 0.0389 | 3.6816 | 3.6816 | 3.6816 | 3.6816 | 3.6816 |
| Random | 0.0399 | 0.0397 | 0.0396 | 0.0396 | 0.0396 | 3.2824 | 3.5017 | 3.6084 | 3.6468 | 3.6487 |
| AR2N | 0.0397 | 0.0397 | 0.0397 | 0.0397 | 0.0397 | 3.6610 | 3.6610 | 3.6610 | 3.6610 | 3.6610 |
| GF | 0.0402 | 0.0394 | 0.0394 | 0.0394 | 0.0394 | 3.6453 | 3.6453 | 3.6453 | 3.6453 | 3.6453 |
| GB | 0.0392 | 0.0392 | 0.0392 | 0.0392 | 0.0392 | 3.6610 | 3.6610 | 3.6610 | 3.6610 | 3.6610 |
| GL | 0.0412 | 0.0406 | 0.0406 | 0.0406 | 0.0406 | 2.2905 | 2.8499 | 3.0970 | 3.0989 | 3.1171 |
| EA | 0.0404 | 0.0400 | 0.0398 | 0.0395 | 0.0395 | 2.6709 | 3.3097 | 3.4353 | 3.4391 | 3.4432 |
| BO | 0.0399 | 0.0396 | 0.0394 | 0.0392 | 0.0392 | 3.0748 | 3.2248 | 3.4774 | 3.6241 | 3.6759 |
| RL | 0.0411 | 0.0406 | 0.0405 | 0.0403 | 0.0403 | 2.5915 | 2.8723 | 3.2453 | 3.2475 | 3.3514 |
| LLM | 0.0397 | 0.0397 | 0.0397 | 0.0397 | 0.0397 | 3.6610 | 3.6610 | 3.6610 | 3.6610 | 3.6610 |
| LLM-CoT | 0.0394 | 0.0394 | 0.0394 | 0.0394 | 0.0394 | 3.6610 | 3.6634 | 3.6655 | 3.6655 | 3.6679 |

Table 6: Performance of ten RDB-to-graph modeling methods on the `rel-stack` dataset (left) and the `rel-trial` dataset (right) under varying budget levels.

| Task Name | post-post-related (**MAP (%)** ↑) | | | | | study-outcome (**AUC-ROC (%)** ↑) | | | | |
|---|---|---|---|---|---|---|---|---|---|---|
| **Methods** | **Budget (%)** | | | | | **Budget (%)** | | | | |
| | **1%** | **2%** | **3%** | **4%** | **5%** | **1%** | **2%** | **3%** | **4%** | **5%** |
| Best | 12.040 | 12.040 | 12.040 | 12.040 | 12.040 | 70.913 | 70.913 | 70.913 | 70.913 | 70.913 |
| Random | 11.751 | 11.803 | 11.835 | 11.867 | 11.900 | 70.178 | 70.338 | 70.372 | 70.375 | 70.407 |
| AR2N | 10.823 | 10.823 | 10.823 | 10.823 | 10.823 | 68.091 | 68.091 | 68.091 | 68.091 | 68.091 |
| GF | 11.903 | 11.903 | 11.903 | 11.903 | 11.903 | 69.572 | 69.572 | 69.572 | 69.572 | 69.572 |
| GB | 11.165 | 11.165 | 11.165 | 11.165 | 11.165 | 69.413 | 69.413 | 69.413 | 69.413 | 69.413 |
| GL | 10.555 | 10.555 | 10.555 | 10.555 | 10.555 | 69.490 | 69.490 | 69.490 | 69.490 | 69.490 |
| EA | 11.786 | 11.897 | 11.946 | 11.954 | 11.954 | 70.639 | 70.771 | 70.772 | 70.772 | 70.772 |
| BO | 11.860 | 11.888 | 11.888 | 11.888 | 11.888 | 70.376 | 70.376 | 70.376 | 70.376 | 70.376 |
| RL | 11.706 | 11.776 | 11.826 | 11.863 | 11.863 | 69.959 | 69.959 | 69.959 | 69.959 | 69.959 |
| LLM | 11.111 | 11.386 | 11.386 | 11.386 | 11.386 | 68.556 | 68.556 | 68.556 | 68.556 | 68.556 |
| LLM-CoT | 10.964 | 10.964 | 10.964 | 10.964 | 10.964 | 68.608 | 68.608 | 68.608 | 68.608 | 68.608 |

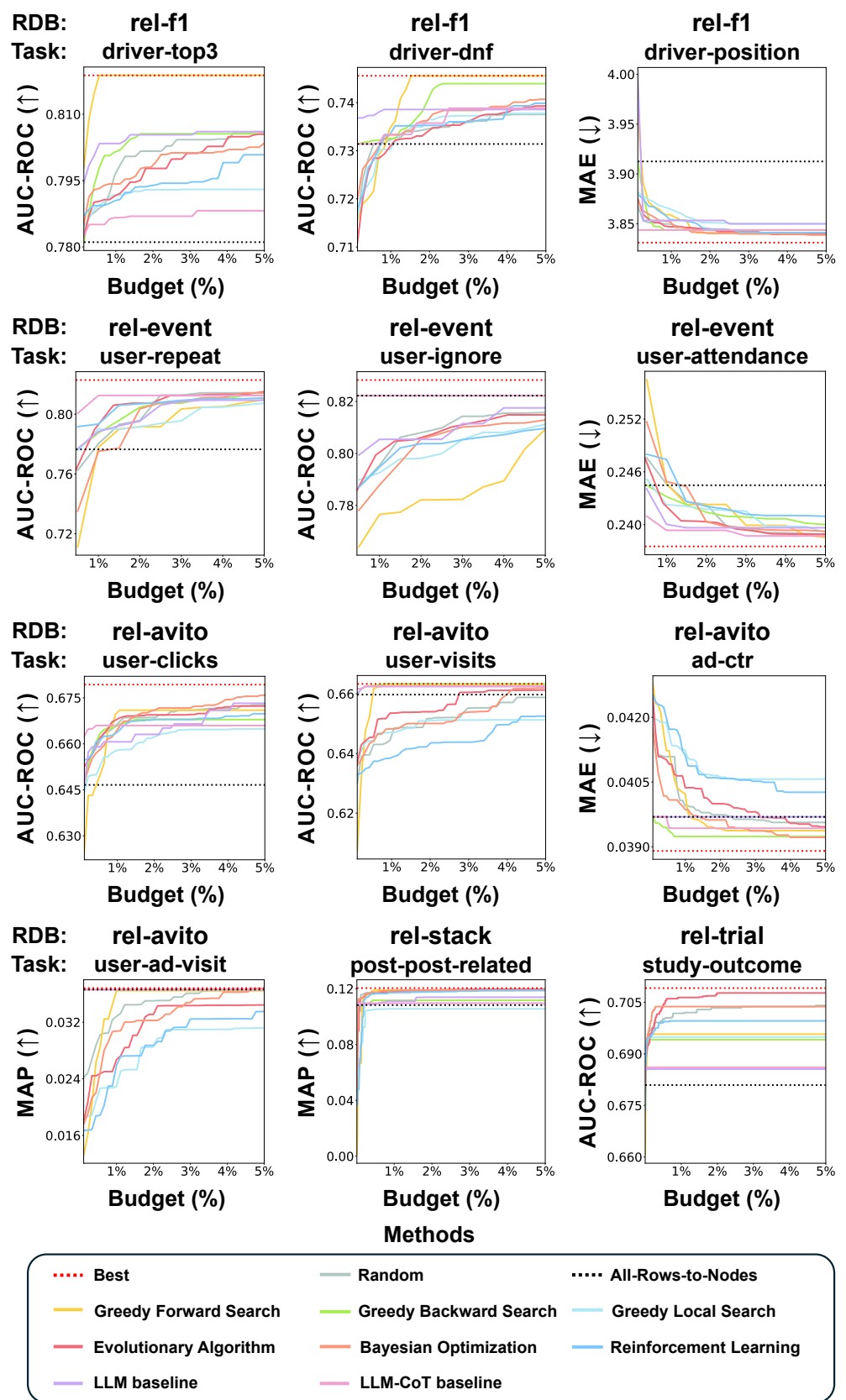

Figure 26: Performance details of ten RDB-to-graph modeling methods on each of the 12 predictive tasks. The performances are computed under varying budget levels, corresponding to the number of graph models evaluated.

## D.2 Analysis of Evaluation Time

In this section, we present a detailed analysis of evaluation time costs for each RDB-to-graph modeling method, which complements the results summarized in Figure 7b in the main paper. Details are provided in Table 7.

Table 7: Analysis of evaluation costs for each method. Total time is computed as the sum of evaluation time and run time, encompassing on-the-fly evaluation and search method costs. All times are averaged over 10 runs.

| Method | Total Time (s) | Run Time (s) | Ratio (Total/Run) |
|---|---|---|---|
| Random | 319016.09 | 0.59 | 540705.24 |
| Greedy Forward Search (GF) | 104246.09 | 0.39 | 267297.67 |
| Greedy Backward Search (GB) | 216032.31 | 0.51 | 423592.76 |
| Greedy Local Search (GL) | 284439.64 | 0.34 | 836587.18 |
| Evolutionary Algorithm (EA) | 696925.57 | 132.62 | 5255.06 |
| Bayesian Optimization (BO) | 595853.52 | 1244.17 | 478.92 |
| Reinforcement Learning (RL) | 753865.31 | 17.66 | 42687.73 |
| LLM baseline (LLM) | 50861.62 | 3588.22 | 14.17 |
| LLM-CoT baseline (CoT) | 28847.97 | 2924.82 | 9.86 |

## D.3 Analysis of LLM Interpretability

Our analysis shows that interpretability can be enhanced when the LLM-CoT baseline provides explicit reasoning traces for each selected graph transformation action. During the action planning phase, we prompted the LLM to explain its rationale behind each operation, allowing us to capture structured insights about how it interprets temporal and structural dependencies.

As shown below, the LLM did not merely select from predefined actions but offered justifications referencing temporal order and causal consistency. These intuitive rationales demonstrate that the LLM's reasoning process is consistent with the underlying graph structure, helping us understand why each transformation was made.

```
<reasoning>
1. Current State Analysis:
- The last action restored the FK-PK relationship between results and races, which
    improved performance slightly
- This suggests that temporal relationships between tables are important for DNF
    prediction
- The key tables for DNF prediction appear to be results, qualifying, races, and
    drivers

2. Examining Remaining Relationships:
- The FK relationship between standings and drivers was removed
- A driver's past performance and championship position could correlate with their
    risk-taking behavior or car reliability

3. Proposed Action:
- Restore the FK-PK relationship between standings and drivers
- Rationale:
  * Driver standings contain cumulative performance metrics (points, position, wins)
  * The temporal nature of standings data could help capture driver form and
      reliability trends
  * Similar to how restoring race relationships improved performance, driver
      standings relationship might capture important patterns

4. Expected Impact:
- Should improve model's ability to capture driver-specific patterns over time
- Aligns with the positive results seen from restoring other temporal relationships

Therefore, I recommend using add_fk_pk_edge to restore the relationship between
    standings and drivers.
</reasoning>
```

# E Further Discussions

## E.1 Limitations

**Graph Design Space Constraints.** As described in Section 3.1 of the main paper, we define the graph design space based on the choices in selecting foreign key relations and modeling rows as nodes or edges. However, additional potential design choices, such as creating dummy tables [35] and advanced feature engineering techniques, are currently excluded due to computational limitations. Expanding the graph design space can present a promising direction for future work.

**GNN Dependence.** Currently, `RDB2G-Bench` provides precomputed evaluations only for four GNNs: GraphSAGE (sum aggregation), GraphSAGE (mean aggregation), GIN, and GPS. Yet, our performance generalization analysis suggests that our findings and the benchmark's utility extend beyond these specific GNNs.

**Limited Task Coverage.** While our original goal is to cover all tasks available in RelBench [29], we have excluded some tasks from `rel-stack` and `rel-trial` due to extremely large graph design spaces and high computational costs. Future work may extend our dataset by covering more tasks.

## E.2 Broader Impact

Our analysis using `RDB2G-Bench` confirms the importance of strategic graph modeling for relational databases, highlighting its academic and practical significance.

From a research perspective, by providing a unified and well-structured set of benchmarks, `RDB2G-Bench` significantly reduces the evaluation cost for researchers and facilitates reproducible comparisons across methods. This enables more efficient and robust validation, thereby accelerating progress in RDB-to-graph modeling research.

Furthermore, leveraging `RDB2G-Bench` can significantly enhance RDB-to-graph modeling methods, enabling industries such as finance, healthcare, and e-commerce to improve efficiency and predictive performance on critical tasks. For example, the financial sector can enhance fraud detection accuracy, while the healthcare sector may develop more precise models for predicting patient outcomes, both directly benefiting from optimized graph models.

## E.3 Analysis of GNN Depth

Following the default configuration of RelBench [29], we fixed the number of GNN layers to two in all experiments. This configuration provides a consistent and computationally efficient setup, and our ablation study further confirms that it is also empirically well-justified. As shown in Tables 8-9, the 2-layer GNN achieves the best average performance across the tasks while also exhibiting a strong correlation with the 3-layer GNN in terms of the top and bottom ranked graph configurations.

We adopt this configuration as it offers a consistent and efficient experimental setup. However, deeper GNNs could potentially better exploit longer relational paths, especially in databases with large-radius schemas. Exploring such depth variations remains a promising direction for future work.

Table 8: Average performance with different GNN depths (1–3 layers).

| RDB | Task Name (Metric) | 1-layer | 2-layer | 3-layer |
|-----|--------------------|---------|---------|---------|
| rel-f1 | driver-dnf (AUC-ROC ↑) | 71.05 ± 0.98 | **71.76 ± 1.39** | 70.87 ± 1.10 |
| rel-avito | user-clicks (AUC-ROC ↑) | 64.03 ± 0.85 | **64.96 ± 1.03** | 64.88 ± 1.34 |
| rel-event | user-attendance (MAE ↓) | **0.249 ± 0.011** | **0.249 ± 0.011** | 0.251 ± 0.010 |

Table 9: Correlation between 2-layer and 3-layer GNN configurations (top and bottom 10%).

| RDB | Task Name | Pearson | Spearman |
|-----|-----------|---------|----------|
| rel-f1 | driver-dnf | 0.851 | 0.816 |
| rel-avito | user-clicks | 0.852 | 0.805 |
| rel-event | user-attendance | 0.963 | 0.883 |

# F    Prompt Design

In this section, we describe the prompts used for our LLM-based baseline implementation, introduced in Section 5.1.

---

**Prompt** Template

**"system" :**
Imagine you are an expert graph data scientist.
**"user" :**
You are expected to construct graph schema based on the original inputs.
 will be given an original schema represented in the dictionary format:
<data>
1. dataset_name: name of the dataset
2. tables: meta data for list of tables, each one will present following attributes
1. name: table name
2. columns: list of columns, each column will have following attributes
  1. name: column name
  2. dtype: column type, can be either text, categorical, float, primary_key, foreign_key, or multi_category. primary_key and foreign_key are two special types of categorical columns, which presents a structural relationship with other tables. Multi_category means this column is of list type, and each cell main contains a list of categorical values. After a column is set as primary_key or foreign_key, it should not be changed to other types.
  3. link_to (optional): if this column is a foreign key, point to which primary key from which table
3. statistics of the table: statistics of the column value of tables. These statistics can be used to help you determine the characteristics of the columns.
</data>

Here are the documents of the actions:
{action document}
{error feedback}

Now, you need to:
1. Actively think about which actions (from the list below) should be conducted to improve the schema.
2. Output all actions you can think of from the above list to make the schema better, and output your selections in the following format:
<selection>
[
  {{"explanation": <explanation for the selection>,
    "action": <selected action>,
    "parameters": <parameters for the action>}},
  {{"explanation": <explanation for the selection>,
    "action": <selected action>,
    "parameters": <parameters for the action>}},
...
]
</selection>
If multiple actions are needed, please list all of them.

<input>
<dataset_stats>
{data statistics}
</dataset_stats>
<task>
{task description}
</task>
<schema>
{graph schema}
</schema>
</input>

{performance feedback}
Note that the current schema may not be optimal, so other actions may yield better results.
Please only halt the program with `None` if you believe no further actions are worth trying.
You can try {budget} more times to improve the performance.
Return your output in the json format inside <selection></selection>.

---

Figure 27: Prompt template for main process.

---
**Action Document** Template

---

Here is the introduction of **add_fk_edge**:
**Description:**
Creates a directed edge from one table to another by adding a foreign key (FK) to primary key (PK) relationship.
Use when you need to represent an important directional relationship between two tables in your graph schema.
**Parameters:**
    from_table_name: the name of the table containing the foreign key
    from_col_name: the name of the foreign key column in to_table
    to_table_name: the name of the table containing the primary key
**Note:** Only the following set of fk_edge can be added: {list of available action parameters}
*or* **Note:** There are no fk_edge that can be added in current schema.

Here is the introduction of **remove_fk_edge**:
**Description:**
Eliminates a directed edge between tables by removing a FK-PK relationship.
Use when a previously modeled relationship doesn't add meaningful context to your graph structure and should be excluded.
**Parameters:**
    from_table_name: the name of the table containing the foreign key
    from_col_name: the name of the primary key column in to_table
    to_table_name: the name of the table containing the primary key
**Note:** Only the following set of fk_edge can be removed: {list of available action parameters}
*or* **Note:** There are no fk_edge that can be removed in current schema.

Here is the introduction of **convert_row_to_edge**:
**Description:**
Transforms what was originally modeled as an entity table into a relationship edge in your graph.
Use when an intermediate table (denoted as edge_table_name) better represents a relationship property between two tables (denoted as table_1_name and table_2_name) rather than being an independent entity.
Note that table_1_name and table_2_name can be equal when the edge_table_name has 2 foreign keys which refer to the same primary key.
**Parameters**:
    table_1_name: the name of the first row table
    table_2_name: the name of the second row table
    edge_table_name: the name of the table to convert to edge between table_1_name and table_2_name
**Note:** Only the following set of edges can be converted from row to edge : {list of available action parameters}
*or* **Note:** There are no edges that can be converted from row to edge in current schema.

Here is the introduction of **convert_edge_to_row**:
**Description:**
Transforms what was modeled as a relationship edge into a proper entity table in your graph.
Use when an edge contains sufficient attributes and identity to justify becoming an entity table with its own properties.
Note that table_1_name and table_2_name can be equal when the edge_table_name has 2 foreign keys which refer to the same primary key.
**Parameters:**
    table_1_name: the name of the first row table
    table_2_name: the name of the second row table
    edge_table_name: the name of the edge table to convert to rows between table_1_name and table_2_name
**Note:** Only the following set of edges can be converted from edge to row : {list of available action parameters}
*or* **Note:** There are no edges that can be converted from edge to row in current schema.

---

Figure 28: Prompt template for action document.

---
**Error Feedback** Template

---

**Warning:** The following actions will cause errors:
Action: {action1}
Parameters: {parameter1}
Error: {error message1}
e.g., Given edge type ({edge table name}) between {table1 name} and {table2 name} is an invalid edge type.
Action: {action2}
Parameters: {parameter2}
Error: {error message2}
e.g., Given edge type ({edge table name}) between {table1 name} and {table2 name} is already connected.
…

---

Figure 29: Prompt template for error feedback.

---

**Performance Feedback** Template

In history actions, after the last {number of last action} actions,
the score has changed from {past performance} to {current performance}.
Since a lower *or* higher score is better, the performance has improved *or* decreased.
*(If decreased)* Please consider either reversing the previous action or exploring alternative actions to improve the schema

---

Figure 30: Prompt template for performance feedback.

---

**Chain-of-Thought Reasoning** Template

Think step by step about whether any of the available actions should be conducted to improve the schema performance.
Consider the history of actions taken, the current score feedback, and potential areas for improvement.

Provide your reasoning in the following format:
**<reasoning></reasoning>**

---

Figure 31: Prompt template for Chain-of-Thought reasoning.

