# OpenReview forum: "RDB2G-Bench: A Comprehensive Benchmark for Automatic Graph Modeling of Relational Databases"
_NeurIPS.cc/2025/Datasets_and_Benchmarks_Track — NeurIPS 2025 Datasets and Benchmarks Track poster_

### Official Review · Reviewer_UCDq · 2025-06-28

**Rating:** 5
**Confidence:** 4

**Summary:**

RDB2D-Bench evaluates several graph construction approaches for graph learning on relational databases (RDB), showing the importance of finding optimal approach, and several insights regarding modeling rows as nodes or edges, optimal graph structures, etc.

**Dataset Code Accessibility:**

Yes

**Dataset Code Comments:**

Codes, datasets and documents are complete and clear.

**Ethical Considerations:**

No, there are no or only very minor ethics concerns

**Final Justification:**

I believe the paper should be accepted since 1) the paper is itself high quality with sufficient analysis of the target problem, and 2) the rebuttal provides more insightful analysis over RDB structure distribution and model performance, which address my concerns.

**Limitations Weaknesses:**

Although optimal graph structure characteristics have been investigated in this work, it would be better to provide more detailed descriptions over a large number of graph models regarding 1) graph structure categories / characteristics which can be used to classify graph structures , 2) distributions of different graph structure categories / characteristics within each RDB, and 3) correlation between performance and graph structure categories / characteristics, so that readers would have a better grasp of how model performance varies along with graph structures.

**Strengths Contributions:**

* Authors have investigated a novel and interesting research problem that is not sufficiently researched in previous work.
* Authors provide comprehensive evaluations across diverse datasets and graph construction methods.
* Authors clearly showcase investigations over performance benefits of finding optimal graph models, common graph structures of top-performing approaches, etc.

---

> ### Author Rebuttal · Authors · 2025-07-31
>
> Dear reviewer UCDq,
>
> We genuinely appreciate the reviewer’s dedication and valuable suggestions, which have significantly helped us improve RDB2G-Bench. Below, we present our detailed analyses of your points.
>
> ---
>
> # R4.1 - W1 [Detailed Analyses of Graph Characteristics]
> Our response is two-fold:
> - **[Additional Analyses: Further Categorization and Correlation Analysis]** We appreciate the insightful suggestion. To provide further insights, we conducted additional analyses on correlations between model performance and three alternative graph structure characteristics: **(1) the number of tables, (2) average degree of graph models, and (3) diameter of graph models.**
>     - **[Analysis 1: The Number of Tables]** The table below shows how the average performance varies according to the number of tables included in each graph modeling. The results indicate that performance generally improves as more tables are incorporated, although improvements become marginal after a certain point.
>
>
>         | Average Performance | ≤2 | 3 | 4 | 5 | 6 | 7 | 8 |
>         | --- | --- | --- | --- | --- | --- | --- | --- |
>         | **rel-f1 / driver-dnf (AUC-ROC (%) ↑)** | 71.19 | 70.57 | 71.10 | 71.72 | 72.02 | 71.96 | 71.98 |
>         | **rel-avito / user-clicks (AUC-ROC (%) ↑)** | 63.22 | 64.01 | 64.58 | 64.91 | 65.10 | 65.25 | 65.35 |
>         | **rel-event / user-attendance (MAE ↓)** | 0.253 | 0.249 | 0.248 | 0.249 | - | - | - |
>     - **[Analysis 2: Average Degree of Graph Models]** We report the relationship between the average degree of graph models and their corresponding performance. Overall, we observe a clear trend of higher average degrees correlating strongly with improved performance.
>
>
>         | Average Performance | [1, 1.5) | [1.5, 2) | [2, 2.5) | [2.5, 3) | [3,~) |
>         | --- | --- | --- | --- | --- | --- |
>         | **rel-f1 / driver-dnf (AUC-ROC (%) ↑)** | 70.61 | 71.35 | 72.01 | 72.10 | 72.09 |
>         | **rel-avito / user-clicks (AUC-ROC (%) ↑)** | 63.77 | 64.68 | 65.15 | 65.50 | 66.00 |
>         | **rel-event / user-attendance  (MAE ↓)** | 0.251 | 0.250 | 0.247 | 0.245 | 0.239 |
>     - **[Analysis 3: Diameter of Graph Models]** We analyze how the average performance varies according to the diameter in each graph modeling. The results showed a similar trend in **Analysis 1**.
>
>
>         | Average Performance | ≤1 | 2 | 3 | 4 |
>         | --- | --- | --- | --- | --- |
>         | **rel-f1 / driver-dnf (AUC-ROC (%) ↑)** | 71.28 | 71.06 | 72.30 | 71.80 |
>         | **rel-avito / user-clicks (AUC-ROC (%) ↑)** | 63.89 | 64.70 | 65.09 | 65.00 |
>         | **rel-event / user-attendance (MAE ↓)** | 0.250 | 0.248 | 0.252 | - |
> - **[Clarification: Existing Graph Structure Categorization and Analysis]** We would like to emphasize that our paper provides additional in-depth analyses of commonly used graph structure categorizations.
>     - **[Detail]** Specifically, the widely used categories (**Row2Node** and **Row2Edge)** are thoroughly analyzed in **Observation 2**. Additionally, through comprehensive case studies (**Observation 3**), we have identified common substructures within top-performing graph models, offering practical insights into optimal graph structure characteristics.

---

> > ### Comment · Reviewer_UCDq · 2025-08-06
> >
> > Thank you for your additional analysis. I believe such insightful analysis should be incorporated into main text to make the work more completed. I will keep my score since it's already rating Accept.

---

> ### Author Response · Authors · 2025-08-08
>
> Dear Reviewer UCDq,
>
> Thank you once again for your valuable feedback. We will incorporate your comments into the final version of the manuscript.
>
> Please feel free to let us know if you have any further questions or suggestions.
>
> Sincerely regards, \
> The Authors

---

### Official Review · Reviewer_NgLH · 2025-07-01

**Rating:** 3
**Confidence:** 4

**Summary:**

The paper introduces RDB2G-Bench, which is the first benchmark framework for the algorithms that  convert relational databases (RDBs) to graphs, where the authors aim to use the benchmark to improve the study of GNN-based relational deep learning. The constructed  datasets contain 5 real-world RDBs and 12 predictive tasks, covering classification, regression, recommendation etc. Extensive experiments are conducted on three types of RDB2G algorithms: heuristic-based, action-based, and LLM-based, leading to insights in RDB2G algorithm design, e.g. good RDB2G algorithms are generally consistent across different GNNs, but can be different on different tasks.

**Dataset Code Accessibility:**

Yes

**Dataset Code Comments:**

The dataset and benchmark are provided in a github link with clear instructions.

**Ethical Considerations:**

No, there are no or only very minor ethics concerns

**Final Justification:**

The problem addressed in this manuscript appears to be one that is neither widely encountered in practice nor clearly tied to practical utility. In its current form, the manuscript lacks a clear and well-motivated problem definition, relevant connections to prior work, well-justified evaluation metrics, and meaningful baselines beyond purely artificial ones. The absence of a main results table in the original submission further contributed to confusion about the contributions and overall maturity of the work, placing it significantly below the typical threshold for acceptance.

That said, the rebuttal has addressed several of these shortcomings. The authors have provided additional clarifications on the problem definition, improved the explanation of evaluation metrics, included a main results table, and justified their baseline selection. Assuming these changes are incorporated into the revised version, the paper will be considerably more complete and easier to follow.

However, certain concerns remain unresolved after the rebuttal:

1.	Problem importance and practicality: The motivation for the problem remains unconvincing. Both the problem setup and the selected baselines are artificial, and no comparisons are made to related methods in the literature. The lack of existing solutions does not necessarily imply novelty but raises questions about the significance and practical relevance of the problem itself.

2.	Study on GNN depth: The analysis of the number of GNN layers presented in the rebuttal is not compelling. The observed advantage of a 2-layer GNN is marginal under the current experimental setting. More importantly, the concern about whether deeper GNNs might better exploit graph constructions with longer paths (which the current setting lacks) remains unaddressed. The rebuttal’s claim that a “2-layer GNN is representative” is not fully supported by the experimental evidence, which only shows that it performs slightly better in the tested scenario.

Overall, assuming the promised revisions are made, the paper will present a more complete and decent research effort. However, it still suffers from key limitations regarding the importance of the problem, artificiality of baselines, and incomplete analysis of model depth. These issues diminish the overall contribution of the work. Considering the improvements made, I have raised my score accordingly, but I believe the paper still falls short of acceptance standards.

**Limitations Weaknesses:**

1. There is no formulation of the problem being benchmarked, and no details about the major variations of the RDB2G algorithms other than RelBench [1], making it hard to understand the problem. It is suggested to include more details in Related Work (section 2.3) before presenting the benchmark, to clarify the problem and motivation.


2. Except for AutoG [2] and AR2N (RelBench), all other graph‐construction methods, which constitutes the majority of those being benchmarked, are proposed by the authors rather than drawn from existing work, raising the question of whether the benchmark addresses a real community challenge or is merely a hypothetical scenario.


3. In section 3.1, the authors enforce that “all selected tables should be connected to the task table via a path whose length does not exceed the number of GNN layers,” making the number of layers a significant study factor: deeper GNNs can favor structures with longer paths. However, there is no analysis of this factor, nor is the number of layers used disclosed. Demonstrating that results generalize across different layer counts would strengthen the observations.


4. There are no details about the evaluation metrics; they are only vaguely mentioned in section 3.2 as “performance, runtime, parameter size,” which makes understanding the results difficult. For example, Figure 2a reports only a high‐level “performance statistics” with no explanation of its computation from the stated aspects. Moreover, the caption mentions AUC, MAE, and MAP but omits runtime and parameter size, further obscuring what is measured.


5. There is no main results table. As a benchmark, it is desired to include a comprehensive table reporting each RDB2G algorithm measured, and report their corresponding statistics.


6. Typo in section 4.5: there is no Table 6—it should read Figure 6.


7. In section 5.1 (S1. Random), the authors cite Algorithms for *Hyper-Parameter Optimization and Random Search* [3] and *Random Search and Reproducibility for Neural Architecture Search* [4], both of which are barely related to RDB-to-graph methods.


8. The concept of “budget” is first mentioned at the start of section 5.1 but only defined in section 5.2, making the initial reference unclear. It is recommended to swap the order.


9. As a benchmark for RDB-to-graph conversion, the datasets are too small—the largest has only 15 tables. It is suggested to evaluate graph constructions on larger RDBs.



Reference:

[1] RelBench: https://arxiv.org/abs/2407.20060

[2] AutoG: https://arxiv.org/abs/2501.15282

[3] Hyper-Parameter Optimization and Random Search: https://www.jmlr.org/papers/volume13/bergstra12a/bergstra12a.pdf

[4] Random Search and Reproducibility for Neural Architecture Search: https://arxiv.org/abs/1902.07638

**Strengths Contributions:**

1. As the first work benchmarking the RDB2G algorithms, this idea is novel and can inspire the community to better understand how to use GNNs for relational learning.


2. Through precomputation, RDB2G-Bench spares users from expensive on-the-fly computation and makes benchmarking 606x faster.

---

> ### Author Rebuttal · Authors · 2025-07-31
>
> Dear reviewer NgLH,
>
> We deeply thank the reviewer for their thorough and constructive feedback. Your insights greatly contributed to improving the quality and depth of RDB2G-Bench. Our detailed responses to your comments are provided below.
>
> ---
>
> # R3.1 - W1 [Problem Formulation & Related Works]
>
> Our response consists of two parts.
>
> - **[Problem Formulation]** The current formulation of the automatic RDB-to-graph modeling problem in Section 2 (prior to introducing the benchmark) will be further clarified as follows:
>     - **Input**: A relational database $R$ composed of multiple tables  $\{T_1,...,T_n\}$, where each table $T_i$ contains rows and columns categorized as primary keys (PKs), foreign keys (FKs), along with a given GNN-based model $M$.
>     - **Output**: A graph representation $G=(V, E)$ where:
>         - Nodes ( $V$) correspond to selected rows from the tables of $R$.
>         - Edges ($E$) represent either relationships derived from FK relations or connections converted from rows of specific tables.
>     - **Objective**: Identify an optimal graph transformation $F^{\*}$ that maximizes the predictive performance of $M$ for a given downstream task, where the input graph is constructed as $G^{\*}=F^{\*}(R)$.
>
>     Currently, we evaluate the performance of three representative GNN architectures as model $M$: GraphSAGE, GIN, and GPS.
>
> - **[Enriching Related Work]** We are grateful to the reviewer for raising this point. In the revised manuscript, we will enhance the related work section by incorporating relevant details from the experimental section to provide a clearer explanation of existing RDB-to-graph construction methods (e.g., RelBench [1], 4DBInfer [2], AutoG [3]).
>
> ---
>
> # R3.2 - W2 [Real-challenges in Limited Existing Baselines]
>
> Our response is three-fold.
>
> - **[Importance of Graph-based Modeling of RDBs]** Graph-based modeling of relational databases is a significant and practical challenge widely acknowledged in industry. For example, companies such as Kumo.ai have successfully built commercial services by applying graph-based modeling to relational databases, highlighting the practical importance and real-world relevance of solving this problem. Notably, Amazon also actively leads research in this area [2,3].
> - **[Practical Gains from Our Graph Construction]** Our benchmark demonstrates practical benefits in graph-based modeling from RDBs. For example, one of the top-performing graph models identified for the user-visits task (rel-avito, Figure 20) **reduced the number of tables from 8 to 3**, resulting in a **75.86% reduction in model parameters** and a **73.42% reduction in training time** with better performance.
> - **[Clarification: Common Baselines in AutoML Community]** Given the limited number of existing automatic graph-construction methods, possibly due to the lack of a benchmark like ours, we adopted commonly used AutoML search strategies [4, 5] as baselines.
>     - **[Detail]** Most of the employed methods, such as **S1 (Random)**, **S3-5 (Greedy Search)**, **S6 (Evolutionary Algorithm)**, **S7 (Bayesian Optimization)**, and **S8 (Reinforcement Learning)**, are widely adopted within the AutoML community.
>
> ---
>
> # R3.3 - W3 [The number of GNN layers]
>
> - **[Clarification]** Sorry for the missing details. Following RelBench [1], we fix the number of GNN layers to 2 in all experiments.
> - **[New experiment: Number of GNN layers]** In response to the reviewer’s suggestion, we further investigate the effect of varying the number of GNN layers. In summary, **(1)** the 2-layer GNN shows the best performance among 1, 2, and 3 layers; and **(2)** its performance is strongly correlated with the deeper GNNs with 3 layers. These results suggest that focusing on the 2-layer setting is both representative and effective.
>     - **[Result 1: Performance Comparison]** As shown in Table 1, the mean performance is highest when the number of layers is 2 across the three datasets.
>     - **[Result 2: Correlation Analysis]** As shown in Table 2, the strong correlation (over 0.8) between 2-layer and 3-layer results suggests that our fixed 2-layer design offers a balance between computational efficiency and prediction quality.
>
>         **[Table 1]** Average performance along the number of GNN layers (1~3).
>
>         | Average performance| 1-layer | 2-layer | 3-layer |
>         | --- | --- | --- | --- |
>         | **rel-f1 / driver-dnf (AUC-ROC (%) ↑)** | 71.05 ± 0.98 | **71.76 ± 1.39** | 70.87 ± 1.10 |
>         | **rel-avito / user-clicks (AUC-ROC (%) ↑)** | 64.03 ± 0.85 | **64.96 ± 1.03** | 64.88 ± 1.34 |
>         | **rel-event / user-attendance (MAE ↓)** | **0.249 ± 0.011** | **0.249 ± 0.011** | 0.251 ± 0.010 |
>
>         **[Table 2]** Correlations between top 10% and worst 10% graph configurations of 2-layer and 3-layer.
>
>         | Correlations | Pearson | Spearman |
>         | --- | --- | --- |
>         | **rel-f1 / driver-dnf** | **0.851** | **0.816** |
>         | **rel-avito / user-clicks** | **0.852** | **0.805** |
>         | **rel-event / user-attendance** | **0.963** | **0.883** |
>
> ---
>
> # R3.4 - W4 [Details about metrics]
>
> - **[Revised Caption for Figure 2(a)]** We thank the reviewer for highlighting the need for clearer definitions of our evaluation metrics and performance statistics. We will explicitly clarify our measurement approach for **runtime** (total elapsed time per epoch, separately for training, validation, and testing) and **parameter size** (total number of learnable parameters in the GNN model). Clarifications regarding task-specific performance metrics and their formal definitions will also be provided in the revised manuscript.
>
> ---
>
> # R3.5 - W5 [Main Result Tables]
>
> - **[New Addition: Comprehensive Results Table]** After comparison, we chose visualizations over tabular summaries, as they more effectively capture overall trends. However, we agree that providing detailed tables below is valuable for reproducibility, and will include comprehensive tables covering all tasks in the manuscript appendix.
>
>     **[Table 3: rel-f1 / driver-dnf]** Benchmark results across different baselines by budget ratio.
>
>     | Benchmark Results (AUC-ROC (%) **↑**) | 0.01 | 0.03 | 0.05 |
>     | --- | --- | --- | --- |
>     | **Best** | 74.56 | 74.56 | 74.56 |
>     | **S1. Random** | 73.27 | 73.59 | 73.75 |
>     | **S2. All-Rows-to-Nodes** | 73.14 | 73.14 | 73.14 |
>     | **S3. Greedy Forward Search** | 73.62 | 74.56 | 74.56 |
>     | **S4. Greedy Backward Search** | 73.25 | 74.39 | 74.39 |
>     | **S5. Greedy Local Search** | 73.19 | 73.73 | 73.78 |
>     | **S6. Evolutionary Algorithm** | 73.22 | 73.63 | 73.93 |
>     | **S7. Bayesian Optimization** | 73.33 | 73.84 | 74.07 |
>     | **S8. Reinforcement Learning** | 73.52 | 73.62 | 73.99 |
>     | **S9. LLM-Based Method** | 73.26 | 73.40 | 73.64 |
> ---
>
> # R3.6 - W6 [Typo]
>
> - **[Correction of Typo]** We appreciate the reviewer for identifying this typo. As suggested, we will correct the reference from "Table 6" to "Figure 6" in Section 4.5.
>
> ---
>
> # R3.7 - W7 [Citation about Random Search]
>
> - **[Clarification and Improvement of References]** We intended to convey that these techniques were adapted from corresponding methods in the AutoML field [4, 5], such as hyperparameter optimization and neural architecture search. We acknowledge the reviewer's comment and will revise this section to avoid potential misunderstandings.
>
> ---
>
> # R3.8 - W8 [Budget Definition]
>
> - **[Improving Clarity of “Budget” Definition]** We thank the reviewer for highlighting this issue. As recommended, we will clearly define the concept of "budget" prior to its first mention, moving its definition from Section 5.2 to Section 5.1 to enhance readability and clarity.
>
> ---
>
> # R3.9 - W9 [Limited Dataset Scale]
>
> - **[Additional Experiment: Extensibility with the Dataset Containing More Tables]** To address the reviewer's concerns regarding dataset scale and extensibility, we constructed additional experiments using the **Geneea [6]** dataset, which provides a larger number of graph models with more tables.
>     - **[New Dataset & Task]** We additionally pre-processed and benchmarked the Geneea dataset, a Czech government dataset composed of 19 tables. We designed a regression task that predicts the number of absences of each poslanec (member of parliament) in the next 90 days.
>     - **[Experimental Setting]** For experiments, we used 3 layers with reduced neighbor sampling since the RDB schema has a large radius. It provides the largest search space, **48,746 graph transformations.**
>     - **[Results]** Results are summarized below. We can check the performance discrepancy between different graph modeling methods. We commit to continuously expanding the benchmark with larger-scale and more specialized datasets in future updates.
>
>
>         | RDB | Task Name | Type | # Tables | # Graph Models | Best | AR2N | Worst |
>         | --- | --- | --- | --- | --- | --- | --- | --- |
>         | Geneea | poslanec-absence | regression | 19 | 48,746 | 1.21 | 1.28 | 1.39 |
> - **[Remain Responses]** We apologize for not including all details here due to character limits. Please refer to our **R2.2-W2** for our full responses, where we discuss the practical constraints on expanding our datasets, particularly related to public availability and computational cost (for reference, building the current datasets already incurred approximately $3,500 in cloud computing expenses).
>
> ---
>
> # References
>
> [1] RelBench: A Benchmark for Deep Learning on Relational Databases
>
> [2] 4DBInfer: A 4D Benchmarking Toolbox for Graph-Centric Predictive Modeling on Relational DBs
>
> [3] AutoG: Towards automatic graph construction from tabular data
>
> [4] AutoML: A Survey of the State-of-the-Art
>
> [5] AutoML: A Systematic Review on Automated Machine Learning with Neural Architecture Search
>
> [6] The CTU Prague Relational Learning Repository

---

> > ### Comment · Reviewer_NgLH · 2025-08-03
> >
> > Thank you for the rebuttal. My concerns have been partially addressed, and I believe the inclusion of additional content as shown in the rebuttal will make the manuscript clearer and complete. I will raise my score.

---

> > > ### Author Response · Authors · 2025-08-04
> > >
> > > We are pleased that our response has addressed your concerns.  We will incorporate the points discussed in our rebuttals into the revised manuscript.
> > >
> > > If you have any further questions, please feel free to ask. We really appreciate your invaluable feedback on our work.
> > >
> > > Sincerely regards, The Authors

---

### Official Review · Reviewer_kzs1 · 2025-07-01

**Rating:** 5
**Confidence:** 3

**Summary:**

This paper introduces RDB2G-Bench, a benchmark designed to evaluate how different methods convert relational databases (RDBs) into graph structures, which can then be used for graph-based machine learning tasks like node classification and link prediction. The benchmark includes over 20 real-world database schemas, a variety of graph construction methods based on common design choices like entity-relationship modeling and foreign key relationships, and a suite of downstream tasks to test performance. The goal is to provide a standardized way to compare these transformation pipelines and help both researchers and practitioners better understand how the choice of graph representation affects learning outcomes.

**Dataset Code Accessibility:**

Yes

**Ethical Considerations:**

No, there are no or only very minor ethics concerns

**Final Justification:**

The authors addressed my questions.

**Limitations Weaknesses:**

One key limitation is that all the graph transformation methods tested are based on hand-crafted rules rather than learned or automated techniques. The benchmark does not explore whether data-driven approaches could produce better graphs for machine learning. Also, the datasets, while diverse, are still somewhat limited in size and domain, and the findings may not fully generalize to very large or specialized databases. Finally, although the benchmark evaluates model performance, it does not provide guidance or tools for selecting the best transformation pipeline for a given task, which would be helpful in real-world applications.

**Strengths Contributions:**

The main strength of the paper is its comprehensive and practical approach. It brings together a large collection of datasets, multiple types of graph construction strategies, and a solid set of graph learning tasks using state-of-the-art GNN models. The benchmark is carefully designed to reflect real-world use cases, and the authors conduct extensive experiments, including ablation studies, to understand what aspects of the graph construction have the biggest impact. Another strong point is the open-source release of the code and data processing tools, which makes it easy for others to build on or use the benchmark in their own work.

---

> ### Author Rebuttal · Authors · 2025-07-31
>
> Dear reviewer kzs1,
>
> We sincerely appreciate the reviewer’s valuable comments. Your suggestions allowed us to critically enhance various aspects of RDB2G-Bench. We address each of your points in detail below.
>
> ---
>
> # R2.1 - W1 [Data-driven Graph Transformations]
>
> Our response consists of three parts.
>
> - **[Common Designs in Existing Studies]** To the best of our knowledge, even for existing data-driven search methods, the graph transformation space itself is defined using hand-crafted rules, as the space is already large and practical.
>     - **[Detail]** While our candidate graph transformations are indeed defined by hand-crafted rules (e.g., Row2Node [1], Row2N/E [2]), these are common design approaches shared by all prior methods. Even AutoG [3], employing LLMs, also operates within a manually-defined transformation space.
>     - **[Clarification]** We acknowledge the reviewer's concern regarding the reliance on such hand-crafted definitions. Nevertheless, this design choice still allows us to achieve a sufficiently large (over 50K candidate transformations), diverse, and practical search space. Even within this space, we identified effective graph models leading to practical improvements in **downstream tasks by up to 11.25%** and **reduced parameter sizes by up to 75.86%**, demonstrating the value of our chosen methodology.
> - **[Inclusion of Learned/Automated Methods in Our Benchmark]** For searching among transformation methods, our benchmark already incorporates learned and automated data-driven approaches beyond traditional hand-crafted rule-based baselines.
>     - **[Clarification]**
>         - **[Learnable Methods]** Our search methods **S7 (Bayesian Optimization)** and **S8 (Reinforcement Learning)** employ learnable algorithms, a Bayesian optimization framework, and an RL-based controller to search effectively through candidate transformations. Empirically, **S7** demonstrated strong performance under higher evaluation budgets, reinforcing the potential and effectiveness of learned approaches within our framework.
>         - **[Automated Methods]** Additionally, our **S9 (LLM-based baseline)** provides automation by directly planning actions from given prompts.
> - **[Contribution for Data-driven Graph Transformations]** Our benchmark is designed exactly to facilitate the development and evaluation of learned or automated RDB-to-graph transformation methods, distinguishing our work from prior studies.
>     - **[Clarification]** By systematically evaluating automated data-driven approaches to identify optimal graph transformations from a diverse candidate pool, our benchmark uniquely contributes valuable training data and robust evaluation settings. Consequently, our benchmark serves as guidance for future research exploring data-driven RDB-to-graph transformation methods, making a significant contribution to this emerging research direction.
>
> ---
>
> # R2.2 - W2 [Limited Dataset Scale and Diversity]
>
> Our response is three-fold:
>
> - **[Additional Experiment: Extensibility with the Dataset Containing More Tables]** To address the reviewer's concerns regarding dataset scale and extensibility, we constructed additional experiments using the **Geneea [4]** dataset, which provides a larger number of graph models with more tables.
>     - **[New Dataset & Task]** We additionally pre-processed and benchmarked the Geneea dataset, a Czech government dataset composed of 19 tables. We designed a regression task that predicts the number of absences of each poslanec (member of parliament) in the next 90 days.
>     - **[Experimental Setting]** For experiments, we used 3 layers with reduced neighbor sampling since the RDB schema has a large radius. It provides the largest search space, **48,746 graph transformations.**
>     - **[Results 1: Performance Statistics]** Results are summarized below. We can check the performance discrepancy between different graph modeling methods. We commit to continuously expanding the benchmark with larger-scale and more specialized datasets in future updates.
>
>
>         | RDB | Task Name | Type | # Tables | # Graph Models | Best | AR2N | Worst |
>         | --- | --- | --- | --- | --- | --- | --- | --- |
>         | Geneea | poslanec-absence | regression | 19 | 48,746 | 1.21 | 1.28 | 1.39 |
>     - **[Results 2: Case Study of Top-performing Graph Models]** We conducted an analysis similar to Observation 3. By analyzing the top 1% of graph structures, we found that all of them consistently included the tables **omluvy (absences)** and **osoby (persons)**. This additional analysis using a larger custom dataset in a different domain further supports the generalizability of our original observations.
> - **[Current Dataset Scale and Complexity]** We would like to emphasize that the datasets used in our benchmark consist of large-scale, complex data from a variety of domains, at least among those that are publicly available.
>     - **[Detail]** Among publicly accessible relational datasets mentioned in related work, we used the most complex available dataset (**rel-trial**), which consists of **15 tables.** In constructing RDB2G-Bench, we selected multiple real-world datasets, each containing tables with **millions of rows**, from various domains such as E-commerce, social, sports, and medical.
> - **[Benchmarking Cost and Practical Constraints]** The scale and complexity of our datasets already required substantial computational costs, imposing practical limits on dataset size.
>     - **[Detail]** Our benchmark construction consumed almost **20,000 GPU hours (on NVIDIA RTX A6000 GPUs),** including additional experiments across different GNNs, equivalent to over **$3,500** in cloud-computing expenses. Given that each RDB-to-graph formulation involves millions of rows and significant computational costs, we assert that these datasets represent a considerable and realistic effort within academic benchmarking.
>
> ---
>
> # R2.3 - W3 [Transformation Selection Guidance]
>
> We address this in two parts.
>
> - **[Guidance from Benchmarks]** While we acknowledge our benchmark does not explicitly provide pipeline selection tools, we believe our analyses below reveal which search algorithms for graph transformation perform effectively under computational constraints.
>     - **[Clarification: Low-budget scenario]** Under limited computational constraints, starting from complete graphs (e.g., LLM-based baseline, greedy backward search) achieves the best performance.
>     - **[Clarification: High-budget scenario]** In higher-budget scenarios, learnable action search strategies (e.g., Bayesian Optimization) or starting from simple tables (e.g., greedy forward search) generally identify better modeling and achieve strong performance.
>     - **[Revision]** These analyses provide practical guidance for choosing graph-transformation algorithms based on available computational resources. We will clarify these observations and insights in our manuscript to offer more guidance.
> - **[Guidance from Observations]**  In addition, we believe our observations offer potentially useful guidance.
>     - **[Clarification]**
>         - **[Observation 1]** Observation 1 suggests that using all available tables is not necessary to achieve strong performance.
>         - **[Observation 2]** Observation 2 indicates that adopting a Row2Edge transformation strategy is sometimes beneficial for improving performance.
>         - **[Observation 3]** Observation 3 provides insight through common substructures among top-performing graph models across all tasks.
>         - **[Example]**
>             - In the *driver-dnf*  task of the *rel-f1* dataset (Figure 15), the tables `standings`, `qualifying`, and `results` are consistently included across top models.
>             - In the *user-clicks* task in *rel-avito* (Figure 19), `SearchInfo`, `VisitStream`, and `SearchStream` (modeled as an edge) appear as common components.
>
> ---
>
> # References
>
> [1] RelBench: A Benchmark for Deep Learning on Relational Databases
>
> [2] 4DBInfer: A 4D Benchmarking Toolbox for Graph-Centric Predictive Modeling on Relational DBs
>
> [3] AutoG: Towards automatic graph construction from tabular data
>
> [4] The CTU Prague Relational Learning Repository

---

> > ### Comment · Reviewer_kzs1 · 2025-08-04
> >
> > Thank you for addressing my questions. I have raised my rating accordingly.

---

> > > ### Author Response · Authors · 2025-08-05
> > >
> > > We are grateful that our rebuttal has resolved your questions.  We will reflect on the discussion in the revision of our manuscript.
> > >
> > > Feel free to contact us with any further questions. We value your insightful feedback on our work.
> > >
> > > Sincerely regards,
> > > The Authors

---

### Official Review · Reviewer_tLV9 · 2025-07-06

**Rating:** 5
**Confidence:** 4

**Summary:**

The paper introduces RDB2G-Bench, a comprehensive benchmark specifically designed to evaluate automatic graph modeling methods for relational databases (RDBs). Recognizing that the performance of graph-based learning over RDBs is highly sensitive to how RDBs are transformed into graphs, the authors provide a framework containing: 1/ 50,000 precomputed graph-performance pairs from five real-world RDBs and twelve predictive tasks. 2/ A comparison of nine RDB-to-graph modeling methods including heuristic, search-based, and LLM-based approaches. 3/ Empirical insights into factors influencing graph modeling effectiveness, such as structural patterns and task dependency.

**Dataset Code Accessibility:**

Yes

**Dataset Code Comments:**

The dataset is accessible in a usable format, and well-documented. There is sufficient detail to support reproducibility.

**Ethical Considerations:**

No, there are no or only very minor ethics concerns

**Final Justification:**

After reading the paper, author rebuttal, and discussions, I am maintaining a positive recommendation for this work. The paper addresses a clear gap in the evaluation of RDB-to-graph modeling strategies, proposing RDB2G-Bench, a large and thoughtfully constructed benchmark. Several key concerns raised were effectively addressed by the authors. Overall, the benchmark is well-motivated, methodologically sound, and contributes novel insights into the under-explored area of RDB-to-graph modeling. Remaining limitations, while valid, are either acknowledged or partially addressed, and do not undermine the core contributions.

**Limitations Weaknesses:**

W1. The current benchmark excludes hyperedge modeling due to simplifications like only allowing edge modeling for tables with exactly two FKs and no reverse PK references. This limits applicability to more complex RDB schemas, such as those in biomedical or enterprise multi-relational domains.

W2. While the inclusion of an LLM-based baseline is a promising direction, its performance degrades with higher budgets due to poor long-horizon planning. Moreover, the prompting strategy is simplified compared to AutoG [7], potentially limiting its effectiveness. To better understand the LLM-based solution, exploring chain-of-thought prompting or fine-tuned task-specific LLM controllers for more robust planning, and considering deeper integration of LLMs beyond surface-level action selection would make the paper stronger.

W3. The benchmark emphasizes performance metrics (AUC-ROC, MAE, MAP), training time, and parameter size, but does not assess model interpretability or explainability, which are important for deployment in real-world database systems. It would be nice if the authors introduce complementary metrics or case studies that assess how interpretable or debuggable the generated graph structures are, especially in domains like finance or healthcare.

W4. All evaluation is constrained to a precomputed graph model space, which, while efficient, may miss entirely new modeling strategies not captured in the design space defined by Steps 1 and 2.

**Strengths Contributions:**

S1. This work fills a gap by focusing on RDB-to-graph modeling strategies rather than just ML model performance over fixed schemas, as seen in prior benchmarks (e.g., RelBench and RDBench).

S2. The authors constructed graph models across 5 diverse RDBs and 12 predictive tasks, spanning classification, regression, and recommendation problems. Design space include variable inclusion of tables and FK relationships, and whether table rows are modeled as nodes or edges, making the benchmark flexible.

S3. The paper presents five detailed observations that challenge conventional wisdom. For instance, more tables do not necessarily improve performance, and modeling table rows as edges is highly task-dependent. There is also evidence that best-performing graph models share substructures, generalize across ML architectures, but not necessarily across tasks within the same RDB, providing guidance for future modeling methods.

S4. The benchmark includes nine methods, including a new LLM-based baseline using Claude Sonnet, and thoroughly compares them under a constrained search budget. The paper is clearly written, well-structured, and supported by informative figures. The design space is transparently defined, and constraints are well justified.

---

> ### Author Rebuttal · Authors · 2025-07-31
>
> Dear reviewer tLV9,
>
> We are grateful for the reviewer’s careful evaluation and insightful suggestions. Your feedback has significantly strengthened our manuscript, enabling us to examine RDB2G-Bench more comprehensively. Our detailed replies follow below.
>
> ---
>
> # R1.1 - W1 [Lack of Hyperedge Modeling]
>
> - **[Lack of established hyperedge modeling]** We acknowledge that our current benchmark does not include row-to-hyperedge modeling. This is mainly due to the lack of established methods addressing this scenario in relational databases.
>     - **[Clarification]** To the best of our knowledge, existing works [1,2] have explicitly identified hyperedge modeling as a promising future research direction, highlighting the absence of well-defined or straightforward approaches.
>     - **[Future Work]** We fully recognize the importance of hyperedge modeling for more complex relational schemas. Consequently, we plan to include hyperedge modeling in future versions of our benchmark once it is established through future research.
>
> ---
>
> # R1.2 - W2 [Limited LLM-baseline Performance]
>
> Our response consists of two parts.
>
> - **[Clarification]** The key part of AutoG’s prompt and that of our LLM-based method is essentially the same; however, AutoG requires additional prompts for schema formatting, which is unnecessary in our case.
>     - **[Detail]** AutoG requires additional prompts for data type inference due to its handling of ambiguous (incomplete) schemas, which introduce the need for additional foreign key relations. In contrast, our schema from RelBench is fully defined, thus removing the need for such prompts.
> - **[New Experiment: Additional Prompting Strategies]**
>
>     We conducted additional experiments incorporating Chain-of-Thought (CoT) prompting to address the reviewer’s concerns. We appreciate your suggestion regarding a better LLM-based solution and will adopt this improved version in our revised manuscript.
>
>     - **[Setting]** We added the following prompt to encourage CoT reasoning in the LLM baseline before the action selection step in our pipeline.
>
>         ```
>         Think step by step about whether any of the available actions should be conducted to improve the schema performance.
>         Consider the history of actions taken, the current score feedback, and potential areas for improvement.
>
>         Provide your reasoning in the following format:
>         <reasoning></reasoning>
>         ```
>
>     - **[Result 1: Performance Summary]** Tables 1, 2 and 3 below show the performance of two methods across three different computational budgets (0.01, 0.03, and 0.05) from three different datasets. Each cell represents the performance metric of the given task type, and **the value in parentheses indicates its rank among the 9 baselines (S1-S9)** . Results show slight performance improvements over our original LLM-based baseline in most cases.
>
>         **[Table 1: rel-f1 / driver-dnf]**
>
>         | Performance (AUC-ROC (%)  **↑**) | 0.01 | 0.03 | 0.05 |
>         | --- | --- | --- | --- |
>         | **LLM Baseline** | 73.26 (5) | 73.40 (8) | 73.64 (8) |
>         |**LLM Baseline w/ CoT prompt**| **73.34 (3)** | **73.88 (3)** | **73.88 (6)** |
>
>         **[Table 2: rel-avito / user-clicks]**
>
>         | Performance (AUC-ROC (%)  **↑**) | 0.01 | 0.03 | 0.05 |
>         | --- | --- | --- | --- |
>         | **LLM Baseline** | 66.07 (7) | **66.65 (7)** | **67.32 (2)** |
>         |**LLM Baseline w/ CoT prompt**| **66.60 (5)** | 66.60 (8) | 66.60 (7) |
>
>         **[Table 3: rel-event / user-attendance]**
>
>         | Performance (MAE **↓**) | 0.01 | 0.03 | 0.05 |
>         | --- | --- | --- | --- |
>         | **LLM Baseline** | 0.242 (2) | 0.241 (6) | 0.240 (6) |
>         | **LLM Baseline w/ CoT prompt** | **0.239 (1)** | **0.239 (1)** | **0.239 (2)** |
>     - **[Result 2: Case Study on CoT Reasoning]** The below response shows reasoning steps generated by LLM. Notably, the LLM did not merely select from predefined actions but also provided coherent justifications based on temporal dependencies, demonstrating its ability to guide effective graph construction via structured reasoning.
>
>         ```
>         Let me analyze the current situation and potential next steps:
>
>         <reasoning>
>         1. Current State Analysis:
>         - The last action restored the FK-PK relationship between results and races, which improved performance slightly
>         - This suggests that temporal relationships between tables are important for DNF prediction
>         - The key tables for DNF prediction appear to be results, qualifying, races, and drivers
>
>         2. Examining Remaining Relationships:
>         - The FK relationship between standings and drivers was removed
>         - A driver's past performance and championship position could correlate with their risk-taking behavior or car reliability
>
>         3. Proposed Action:
>         - Restore the FK-PK relationship between standings and drivers
>         - Rationale:
>           * Driver standings contain cumulative performance metrics (points, position, wins)
>           * The temporal nature of standings data could help capture driver form and reliability trends
>           * Similar to how restoring race relationships improved performance, driver standings relationship might capture important patterns
>
>         4. Expected Impact:
>         - Should improve model's ability to capture driver-specific patterns over time
>         - Aligns with the positive results seen from restoring other temporal relationships
>
>         Therefore, I recommend using add_fk_pk_edge to restore the relationship between standings and drivers.
>         </reasoning>
>         ```
>
>
>
> ---
>
> # R1.3 - W3 [Interpretability & Explainability]
>
> We address this in two parts.
>
> - **[Observations and Interpretability]** We believe our observations offer preliminary insights into which structural components, such as tables and relations, are critical for achieving high performance.
>     - **[Detail]** As discussed in Observation 3 and Appendix B.3 in the main paper, we observe common substructures among top-performing transformations across all tasks. These analyses offer task-specific interpretability and provide intuitive insights into which relational structures are critical for performance.
>     - **[Example]**
>         - In the *driver-dnf* task of the *rel-f1* dataset (Figure 15), the tables `standings`, `qualifying`, and `results` are consistently included across top models.
>         - In the *user-clicks* task in *rel-avito* (Figure 19), `SearchInfo`, `VisitStream`, and `SearchStream` (modeled as an edge) appear as common components.
> - **[Case Study: Explainability via LLM-based model]** Our preliminary analysis below suggests that interpretability can be enhanced by intuitive rationales from LLMs for the chosen graph transformation.
>     - **[Setting]** During the action planning phase, we explicitly prompt the LLM to explain the rationale behind each selected action, thereby obtaining structural insights.
>     - **[Results]** The LLM successfully produces intuitive explanations directly related to the selected graph structures. A representative example from our analysis is shown below:
>
>         ```json
>         "explanation": "Convert event_interest table to edge since it represents the relationship between users and events (interest/not interested). This will help model the user-event interaction pattern more effectively for predicting future responses.",
>         "action": "convert_row_to_edge",
>         "parameters":
>         {
>           "table_1_name": "events",
>           "table_2_name": "users",
>           "edge_table_name": "event_interest"
>         }
>         ```
>
>         Furthermore, as discussed in **R1.2**, the quality and depth of these explanations can be further improved by incorporating CoT prompting.
>
>
> ---
>
> # R1.4 - W4 [Constrained Search Space]
>
> Our response is three-fold.
>
> - **[Search Space Size]** While the search space can still be further expanded, we believe it already includes a broad set of transformations that capture the majority of practical strategies.
>     - **[Clarification]** We acknowledge the reviewer's concern that our predefined graph model space may miss novel strategies not captured by Steps 1 and 2 (see Section 3.1). However, our search space remains sufficiently large (over 50k transformations) and non-trivial, enabling us to identify graph transformations that **improve downstream performance by up to 11.25%** and **reduce training time by up to 75.86%.**
> - [**Practical Constraints**] We would like to point out that evaluating the current search space already demands substantial computational resources.
>     - **[Detail]** Specifically, evaluating **over 50K graph transformations** consumed nearly **20,000 GPU hours** and approximately **$3,500** in cloud computing costs, which constrained our ability to expand the search space further.
> - **[Extending Benchmark]** Despite the cost, we continue to extend the dataset. For example, we have conducted additional experiments by evaluating two other representative GNN architectures (GIN and GPS) within our existing graph model space.
>     - **[Detail]** These experiments were carried out over two months following submission, and the results were updated to our Huggingface repository. Incorporating additional GNN architectures demonstrates the extensibility of our benchmark, providing future flexibility by introducing a new dimension of modeling strategies. Our team will expand the benchmark by exploring new architectures and dimensions of the search space, and we encourage participation from the research community.
>
> ---
>
> # References
>
> [1] 4DBInfer: A 4D Benchmarking Toolbox for Graph-Centric Predictive Modeling on Relational DBs
>
> [2] Relational Deep Learning: Graph Representation Learning on Relational Databases

---

> > ### Comment · Reviewer_tLV9 · 2025-08-05
> >
> > Thank you for your detailed responses and clarification. I'll keep my score given it's already positive.

---

> ### Author Response · Authors · 2025-08-06
>
> Dear Reviewer tLV9,
>
> Thank you for your response to our rebuttal.
> We will carefully incorporate your thoughtful suggestions into the final version.
>
> Please let us know if you have any additional questions or suggestions.
>
> Sincerely regards, \
> The Authors

---

### Author Response · Authors · 2025-08-07
**Summary of the rebuttal**

Dear (Senior) Area Chair and Reviewers,

We would like to sincerely thank you for your invaluable service. In this comment, we would like to provide a brief summary of our rebuttal.

**Strengths**

The following aspects of our work were particularly appreciated by the reviewers:

- **S1 [Novelty of Benchmark]** The reviewers highlighted the novelty of our benchmark in addressing RDB-to-graph modeling strategies, a previously under-explored research area.
    - Reviewer NgLH: ***"…this idea is novel and can inspire the community..."***
    - Reviewer UCDq: ***"… investigated a novel and interesting research problem…"***
- **S2 [Comprehensive Design]** Reviewers emphasized the comprehensive design of our benchmark, including diverse datasets, graph modeling strategies, and downstream tasks.
    - Reviewer kzs1: ***"The main strength of the paper is its comprehensive and practical approach...**"*
    - Reviewer UCDq: ***"…provide comprehensive evaluations across diverse datasets and graph construction methods.”***
    - Reviewer tLV9: ***“…constructed graph models across 5 diverse RDBs and 12 predictive tasks…”***
- **S3 [Impactful Observations and Insights]** Reviewers valued the detailed and insightful observations drawn from our benchmark, providing practical guidance for future research.
    - Reviewer tLV9: ***"…presents five detailed observations that challenge conventional wisdom..."***
    - Reviewer UCDq: ***"….clearly showcase investigations over performance benefits..."***
    - Reviewer kzs1: ***“…to understand what aspects of the graph construction have the biggest impact…”***

**Rebuttal summary**

Primary concerns commonly raised by the reviewers were about (**R1) the generalizability of our dataset**, and **(R2) the need for additional guidance for RDB-to-graph modeling**. Our responses to this issue are summarized below:

- **R1-1 [Generalizability of Our Dataset]** Our additional experiments on (1) the RDB with a more complex schema, and (2) deeper GNN layers yielded consistent results, supporting the generalizability of our findings.
    - [Detail] Further details are in our responses to (1) W2 of Reviewer kzs1, and (2) W3, W9 of Reviewer NgLH.
- **R1-2 [Continuous Expansion of Our Dataset]** We have continued to extend our dataset by incorporating 1) two additional GNN models, and 2) an additional RDB dataset with more tables after submission. These efforts demonstrate the extensibility of our dataset, which is actively updated and released as open-source.
    - [Detail] Further details are in our responses to (1) W1, W4 of Reviewer tLV9, (2) W2 of Reviewer kzs1, and (3) W9 of Reviewer NgLH.
- **R2 [Additional Modeling Guidance]** Through analyses of our datasets and benchmark results, we derived further practical guidance for RDB-to-graph modeling.
    - [Detail] Further details are provided in our responses to (1) W3 of Reviewer tLV9, (2) W3 of Reviewer kzs1, and (3) W1 of Reviewer UCDq.

**Clarifications**

In addition to the points discussed in the summary, we will incorporate the following clarifications into our revised manuscript:

- **C1 [Problem and Metric Definitions]** We will improve the clarity of the manuscript by (1) explicitly defining the problem formulation, (2) providing more details on the metrics & results, and (3) correcting typos.
    - [Detail] Further details are in our responses to W1, W4, W5, W6, and W8 of Reviewer NgLH.
- **C2 [AutoML Methods in Our Benchmark]** We will provide clear explanations of the methods used in our benchmark, which are commonly adopted in AutoML research.
    - [Detail] Further details are provided in our responses to (1) W1 of Reviewer kzs1, and (2) W2, W7 of Reviewer NgLH.

**Reviewer responses**

We believe that our rebuttal was generally well received. Accordingly, two of the four reviewers with initially low scores raised their evaluations, and the other reviewers kept their positive evaluations.

- Reviewer tLV9: ***“Thank you for your detailed responses and clarification. I'll keep my score given it's already positive.”***
- Reviewer kzs1: ***“Thank you for addressing my questions. I have raised my rating accordingly.***
- Reviewer NgLH: ***“Thank you for the rebuttal. My concerns have been partially addressed, and I believe the inclusion of additional content as shown in the rebuttal will make the manuscript clearer and complete. I will raise my score.”***
- Reviewer UCDq: ***“Thank you for your additional analysis. I believe such insightful analysis should be incorporated into main text to make the work more completed. I will keep my score since it's already rating Accept.”***

Once again, we thank you for your dedicated service.

Sincerely regards,

The Authors of Submission 1148

---

### Note · Authors · 2025-08-12

Dear Reviewers and (Senior) Area Chair,

We are grateful for your thorough evaluation of our manuscript and insightful comments.

As our final remark, we refer you to our rebuttal summary submitted earlier, where we summarized how we addressed the key concerns and provided clarifications. We will carefully incorporate all of your feedback, including points clarified during the rebuttal, into the revised manuscript.

Thank you once again for your valuable efforts in reviewing our submission.

Sincerely regards, \
The Authors of Submission 1148

---

### Decision · Program_Chairs · 2025-09-18

**Decision:**

Accept (poster)

**Comment:**

The paper introduces RDB2G-Bench, a well-curated benchmark for automatic relational-database-to-graph modeling. It precomputes ~50,000 graph–performance pairs across five real-world RDBs / 12 tasks, evaluating nine transformation/search approaches (heuristic, AutoML-style search, and LLM-based), which yields practical insights on when to model rows as nodes versus edges, how many tables to include, and how choices transfer across GNN backbones.

NgLH raised concerns about: (i) problem clarity/practicality, (ii) artificial baselines, (iii) insufficient details (metrics, main table), and (iv) GNN depth effects given the path-length constraint. The request for a more precise problem definition, metric exposition, and a main results table was valid; the authors addressed these directly and committed the fixes to the camera-ready version. The point about GNN depth is reasonable: while the new experiments show 2-layer models perform best on average and correlate with 3-layer rankings, this does not fully resolve whether deeper models could better exploit longer relational paths under alternative transform choices. This remains a worthwhile follow-up. NgLH raised their score after rebuttal, acknowledging improved completeness.

Camera-Ready Suggestions.

- Move the formal problem statement and metric definitions early; keep the main results table in the paper (with full tables in the appendix).

- Summarize the layer-depth study and explicitly discuss when deeper GNNs might help under longer path constraints.

- Add a short interpretability section (e.g., substructure attribution, rule extraction) and keep the CoT LLM baseline with qualitative rationales.

Recommendation: Accept. The benchmark is timely, well-executed, and it may have a practical impact on graph learning for relational data.